# Towards Fair Video Summarization

**Anshuman Chhabra**                                    *chhabra@ucdavis.edu*
*University of California, Davis*

**Kartik Patwari**                                      *kpatwari@ucdavis.edu*
*University of California, Davis*

**Chandana Kuntala**[*]                                 *chandanakuntala21@gmail.com*
*Indira Gandhi Delhi Technical University for Women*

**Sristi**[*]                                           *sristi0108@gmail.com*
*Indira Gandhi Delhi Technical University for Women*

**Deepak Kumar Sharma**                                 *dk.sharma1982@yahoo.com*
*Indira Gandhi Delhi Technical University for Women*

**Prasant Mohapatra**                                   *pmohapatra@usf.edu*
*University of South Florida, Tampa*

**Reviewed on OpenReview:** *https://openreview.net/forum?id=Uj6MRfR1P5*

## Abstract

Automated video summarization is a vision task that aims to generate concise summaries of lengthy videos. Recent advancements in deep learning have led to highly performant video summarization models; however, there has been a lack of attention given to fairness and unbiased representation in the generated summaries. To bridge this gap, we introduce and analytically define the fair video summarization problem, and demonstrate its connections to the well-established problem of fair clustering. To facilitate fair model development, we also introduce the *FairVidSum* dataset, which is similar in design to state-of-the-art video summarization datasets such as *TVSum* and *SumMe*, but also includes annotations for sensitive attributes and individuals alongside frame importance scores. Finally, we propose the SumBal metric for quantifying the fairness of an outputted video summary. We conduct extensive experiments to benchmark the fairness of various state-of-the-art video summarization models. Our results highlight the need for better models that balance accuracy and fairness to ensure equitable representation and inclusion in summaries. For completeness, we also provide a novel fair-only baseline, FVS-LP, to showcase the fairness-utility gap models can improve upon.

## 1 Introduction

With the rapid growth of video content on the internet, there is an increasing need to automatically summarize lengthy videos to provide users with a condensed version that contains the most salient information. This has led to the machine learning (ML) vision task of automated video summarization, which entails generating a short, representative summary video (comprised of key-frames) of a longer input video that showcases its main content and events. In recent years, deep learning (DL) based models have achieved the state-of-the-art (SOTA) in video summarization by leveraging powerful feature representations and learning complex relationships between video frames (Apostolidis et al., 2021a). Furthermore, the video summarization task itself is employed in several downstream practical applications, such as surveillance

---

[*]Equal Contribution.

(Senthil Murugan et al., 2018; Thomas et al., 2017), video retrieval (Gong & Liu, 2003; Peng & Ngo, 2006), among others. Advancements in video summarization can then directly impact and improve performance on these downstream video analysis tasks.

The ML/DL community has also recently pivoted to studying model fairness as models can exhibit harmful biases against minority groups and individuals (Mehrabi et al., 2021). These issues of unfairness have been evidenced in many high-impact applications as well.[1] Thus, with the growing use of video summarization in numerous applications, it is extremely important to ensure that these automated methods are fair and unbiased, both at the *individual-level* (Berk et al., 2017) and at the *group-level* (Dwork et al., 2012) (such as with regards to *sensitive attributes* like ethnicity and sex). However, no work has been undertaken in *fair video summarization*, while significant progress has been made in developing fair models for other tasks/fields in ML/DL (Chhabra et al., 2021a; Mehrabi et al., 2021).

To bridge this gap, we propose and analytically define the *fair video summarization problem*, to allow for the development of fair methods at the individual- and group-level. Our fairness definition is conceptualized similar to the well-studied problem of representation-based fair clustering (Chierichetti et al., 2017; Chhabra et al., 2021a). Another hindrance to fairness evaluation in summarization models stems from a lack of any video summarization datasets containing individuals, and appropriate annotations reflecting their sensitive attributes (such as sex and ethnicity). Current benchmark datasets used to train and evaluate video summarization models are the *TVSum* (Song et al., 2015) and

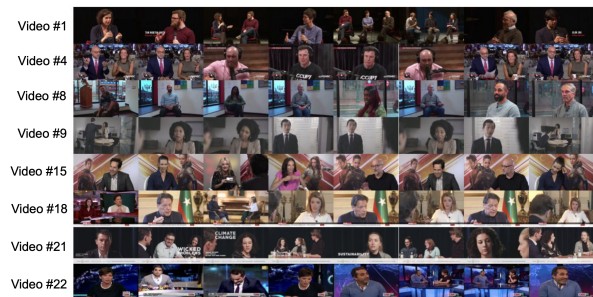

Figure 1: Video samples from *FairVidSum*.

*SumMe* (Gygli et al., 2014) datasets which do not primarily contain human subjects and lack information regarding any protected groups or sensitive attributes. To this end, we propose the *FairVidSum* dataset containing multiple individuals spanning diverse settings such as interviews, podcasts, and panel discussions. Unlike the other datasets, we provide manual annotations for sensitive attributes (fairness) as well as frame importance scores (utility). Furthermore, we also propose novel metrics to evaluate (un)fairness in current SOTA supervised and unsupervised video summarization models and benchmark them. Finally, for completeness we also propose a novel unsupervised method for fair video summarization named FVS-LP, which is a linear program (Schrijver, 1998) based simple baseline that only optimizes for fairness. Finally, we would also like to emphasize that while video summarization has been studied extensively over the past few decades, our work is primarily concerned with more recent *learning-based* video summarization approaches (proposed by the vision community) as they are highly performant (Apostolidis et al., 2021a). For more details on *classical video summarization* studied by the multimedia community, please refer to (Truong & Venkatesh, 2007) for a survey of existing methods. In summary, through this work, we make the following contributions:

- We provide an analytical definition for the *fair video summarization* problem to allow for the development of accurate, fair, and unbiased video summarization models that can help ensure equal representation and inclusion in video content. Frames from a few randomly sampled FairVidSum videos are shown in Figure 1.
- We propose the *SumBal* fairness metric to evaluate model fairness, which is derived from the Balance (Chierichetti et al., 2017) metric proposed to measure fairness in unsupervised learning.
- We introduce the *FairVidSum* benchmark dataset, designed similarly to existing SOTA video summarization benchmarks TVSum (Song et al., 2015) and SumMe (Gygli et al., 2014), which contains annotated individual and group-level fairness information.
- Using *FairVidSum* and SumBal, we benchmark numerous SOTA supervised and unsupervised models for unfairness. We find that since most models do not optimize for fairness, they can be highly unfair, prompting the need for newer methods that can balance both accuracy and fairness.

---

[1]Notable examples include Microsoft's Tay chatbot that became racist and homophobic after training on user data online (Neff, 2016), and the COMPAS tool which recommended that black individuals were more likely to reoffend compared to other ethnicities, despite no statistical differences between the individuals themselves (Angwin et al., 2016).

- For completeness, we also propose FVS-LP: a novel baseline for unsupervised fair video summarization that solely optimizes for fairness, to empirically demonstrate the fairness-utility gap. Based on FVS-LP, we showcase a simple sampling strategy that can control the fairness-utility tradeoff as well.

## 2 Related Work

**Classical Video Summarization.** Video summarization has been studied extensively by the multimedia community over the past few decades (Truong & Venkatesh, 2007). However, these approaches are not learning-based (that is, they do not employ ML/DL models to undertake the video summarization task), and tend to be less performant than learning-based approaches. Moreover, classical approaches often utilize multiple modalities for summarization, whereas learning-based approaches often consider only visual information contained in video frames. For instance, in Ma et al. (2002), the authors propose a simple video summarization method for fusing visual, audio, and linguistic information contained in the video. Other seminal classical summarization approaches include Ngo et al. (2005) where the authors use temporal graph modeling methods; Taskiran et al. (2006) where speech transcripts are used instead of visual/audio information; Chen et al. (2009) which proposes a concept-entity method for summarization; and Yu et al. (2003) which utilizes the annotators' browsing patterns (logs) as an alternative method for future summarization, among others. Note that in this work, we focus on recent learning-based approaches developed by the vision community (Apostolidis et al., 2021a) since they achieve state-of-the-art performance on current benchmarks (*TVSum* and *SumMe*), which we discuss in more detail below.

**Learning-Based Video Summarization.** Video summarization approaches can be categorized (Apostolidis et al., 2021a) as either *supervised* (frame-level importance scores are used in training) (Zhang et al., 2016; 2019; Huang & Wang, 2019; Fu et al., 2019; Lebron Casas & Koblents, 2019) or *unsupervised* (only visual frame information is used during training) (Mahasseni et al., 2017; Yuan et al., 2019; He et al., 2019; Yaliniz & Ikizler-Cinbis, 2021; Apostolidis et al., 2019; Zhou et al., 2018a) with regards to the learning setting, and as *unimodal* (only visual information is used for training) (Zhang et al., 2016; He et al., 2019; Yuan et al., 2019; Fu et al., 2019) or *multimodal* (other video metadata is also utilized) (Otani et al., 2017; Wei et al., 2018; Lei et al., 2018; Zhou et al., 2018b; Yuan et al., 2017; Song et al., 2016) with regards to the input data type. *Unsupervised unimodal* approaches model the application scenarios for video summarization better as annotated frame importance scores and additional video metadata (such as transcripts) are generally hard to obtain (Apostolidis et al., 2021a). For video summarization, the SOTA supervised approaches constitute DSNet (Zhu et al., 2020), PGL-SUM (Apostolidis et al., 2021b), among others and the SOTA unsupervised approaches constitute CA-SUM (Apostolidis et al., 2022), AC-SUM-GAN (Apostolidis et al., 2020a), SUM-GAN-AAE (Apostolidis et al., 2020b), SUM-GAN-SL (Apostolidis et al., 2019). All these models are benchmarked on the *TVSum* (Song et al., 2015) and *SumMe* (Gygli et al., 2014) datasets.

**Fairness in Machine Learning and Summarization.** While video summarization has not yet been studied from the purview of fairness, fair models have been developed for various ML tasks and problem settings (Mehrabi et al., 2021; Chhabra et al., 2021a). These include supervised learning (Agarwal et al., 2018; Zafar et al., 2017), unsupervised learning (Chierichetti et al., 2017; Chhabra et al., 2022; Kleindessner et al., 2019b), recommendation systems (Rastegarpanah et al., 2019; Pitoura et al., 2022), active learning (Anahideh et al., 2022; Shen et al., 2022), outlier detection (Song et al., 2021; Davidson & Ravi, 2020), among others. Works have also investigated the interplay between fairness and other desirable model behaviors, such as robustness (Chhabra et al., 2023; 2021b). Further, fairness has also been studied for data summarization such as for k-center based summarization (Kleindessner et al., 2019a; Chiplunkar et al., 2020; Angelidakis et al., 2022) and text summarization (Shandilya et al., 2018; Keswani & Celis, 2021). However, these approaches are not general– they are highly specific to the learning algorithm being used (for e.g. k-center (Gonzalez, 1985)) and the fairness definitions employed are not consistent with our goal. As such, these are not applicable for fair video summarization. Moreover, fairness can be enforced at the *pre-processing* (before training), *in-processing* (modifying the model), or *post-processing* (after training) stage of the learning pipeline (Mehrabi et al., 2021). As will be clear in subsequent sections, the proposed FVS-LP baseline belongs to the in-processing category, as it enforces fairness constraints during training.

## 3 Problem Statement and Preliminaries

In this section, we first describe the standard video summarization problem and discuss protocols for evaluating utility of trained models. Then, we introduce the fair video summarization problem– we provide an analytical formulation, along with evaluation metrics and motivating use-cases. Note that we only consider unimodal video summarization models in this paper as these are more commonly used in the context of deep learning (Apostolidis et al., 2021a).

### 3.1 The Video Summarization Problem

**Unsupervised Video Summarization.** Let a video $V$ consist of $n$ frames $X = \{x_1, x_2, ..., x_n\}$ where $x_i \in \mathbb{R}^d$. These are sampled at some frequency (usually 2 frames per second (Apostolidis et al., 2021a)) from $V$ and hence, $n$ is generally large. Here, $d$ is the dimension of the feature descriptor of the frame (for example, this could represent features extracted per frame using a ResNet (He et al., 2016)). An *unsupervised* video summarization model can then generally be denoted as $\mathcal{M}^{\text{unsup}}$ that takes in as input a summary length requirement $k \ll n$ and the original video frame set $X$, and outputs a set of key frames constituting the video summary as $S = \{x_j\}_{j=1}^k \subseteq X$. That is, $\mathcal{M}^{\text{unsup}}(X, k) = S$, where $S \in \mathbb{R}^{k \times d}$. The summary length budget $k$ is generally set to be 15% of the original video length, that is, $k/n = 0.15$.

**Supervised Video Summarization.** While unsupervised variants are better suited for video summarization (Apostolidis et al., 2021a) since they model the application scenarios in a more realistic manner (human-level annotations are hard to obtain), supervised models are employed as well. A supervised model also takes in as input $Y = \{y_i\}_{i=1}^m$ where $0 < y_i \leq 1$ is an importance score given by a human annotator for a corresponding frame $x_i \in X$.[2] Annotations are only obtained for a small subset of frames $m$ since $n$ can be quite large. Thus, for a supervised model, we can obtain a summary as $\mathcal{M}^{\text{sup}}(X, k, Y) = S$ where $|S| = k$.

**Evaluating Models.** Trained video summarization models are evaluated based on the agreement of the generated summary for a video with its ground truth summary obtained using the annotated importance scores provided by a given user. Note that obtaining summaries from the importance scores $Y$ is also an optimization problem since we have a budget $k$ for the length of the summary. Usually, the 0/1 knapsack (Martello et al., 1999) problem is used to obtain user summaries in this manner (Zhang et al., 2016). Thus, if we have $u$ users who annotated video $V$, we will have summaries available denoted as $O_1^V, O_2^V, ..., O_u^V$ corresponding to each user. The given model generates a summary $S^V$ for a particular video $V$. We can then obtain the precision and recall between each $O_i^V$ and $S^V$, denoted as $p_i^V$ and $r_i^V$, respectively. To evaluate models, we then calculate the average pairwise $F_\beta$-measure averaged over all user summaries as:

$$F_\beta^V = \frac{1}{u} \sum_{i=1}^u \frac{(1 + \beta^2) \times p_i^V \times r_i^V}{(\beta^2 \times p_i^V) + r_i^V} \tag{1}$$

Usually, $\beta$ is set to 1 (Song et al., 2015), so we compute the average pairwise $F_1^V$-measure for a given video $V$. These values are then averaged over all videos $V$ in the test set, and overall $\bar{F}_1$-measure is calculated. Further, note that for the supervised setting, videos that are used for model training cannot be used in the evaluation/test set. Hence, cross validation is generally undertaken (Apostolidis et al., 2021a) to create 80% (train) - 20% (test) splits. Although this issue of train-test splits does not arise for unsupervised models, for consistency, we follow the same protocol for evaluation of all models.

### 3.2 The Fair Video Summarization Problem

**Problem Statement.** We now define the fair video summarization problem for a video $V$. Here, along with $X$, $Y$, and $k$, we are also given (fairness) information regarding $g$ individuals or protected groups as $\mathcal{H} = \{H^1, H^2, ..., H^g\}$ where $H^j \in \{0,1\}^n$ and $H_i^j = 1$ implies that individual/group $j$ is present in frame $i$. Conversely, $H_j^i = 0$ implies that individual/group $j$ is absent in frame $i$. Note that unlike importance scores these are not subjective decisions, so we have discrete labels indicating individual/group presence

---

[2]Importance score annotations are generally obtained between 1 (least important) and 5 (most important) and then normalized to lie between 0 and 1.

in frames. Note that this abstraction using $H^j$ is very flexible, and can allow for the development of fair models that optimize for individual fairness or group-fairness. For individual fairness this constitutes the idea that all persons in the video should be represented in approximately the same proportions in the generated summary as they appear in the entire video. For group fairness, this could constitute different groups being represented in the same proportions in the summary frames as their proportions in the overall video frames. For example, for *ethnicity* as the sensitive attribute, this would necessitate proportional representation for each ethnicity in summary frames compared to total video frames. This is the very notion of *disparate impact* (Kleinberg et al., 2018) and ensures that no protected group or individual[3] be adversely affected as a result of a predictive algorithm.

A fair video summarization model $\mathcal{M}^{\text{fair}}$ then also takes in as input $\mathcal{H}$ and generates summary $S$ for video $V$ as $\mathcal{M}^{\text{fair}}(X, k, \mathcal{H}) = S$. Along with optimal utility performance, the model must ensure that the proportion of appearance of entities represented by $\mathcal{H}$ are as close as possible to their overall proportions in the video $V$. The supervised fair variant can also be defined similarly. As is evident by our definition, in this work we only consider optimizing one type of $\mathcal{H}$ at a time (that is, *sex* consisting of *male/female* appearances in frames). However, as our dataset has information regarding multiple groups, this can be studied in future work.

**Motivating Examples.** Consider a platform such as YouTube (Covington et al., 2016). For simplicity, consider a set of news/podcast videos on the platform that have one male and one female host. Summaries for these videos are generated on the platform as the user browses the homepage. Here too, if a standard summarization model is used, there is no guarantee that the outputted video summary will respect the appearance proportions of the male/female hosts in the original video. In fact, even if the original video has equal appearance proportions for both male/female hosts, the model might skew these proportions heavily in the generated summary. A *fair* video summarization model instead would ensure that both male and female hosts appear in roughly the same amounts as in the original video, leading to fair representation.

Consider another example of a video surveillance application that utilizes video summarization models in the backend, such as in (Senthil Murugan et al., 2018; Thomas et al., 2017), being used by law enforcement with multiple persons appearing in the video. If a standard video summarization model is used, the generated summary footage might have certain individuals appearing for large segments of the summary and might not reflect their actual proportion of appearance in the overall video. As a result, this might lead to a falsified description of the original footage. On the other hand, if a fair video summarization model is used, the individuals would appear in the summary in the same proportions as in the original video footage, and result in a more *fair* overview. The same arguments can be made with regards to people from different ethnicities or gender appearing in the summary footage and preventing discrimination and bias at the group-level.

**Evaluating Unfairness.** Now that we have described the fair summarization problem, it is important to propose metrics for evaluating the discrepancies in fairness. Our basic goal is to measure whether or not each entity constituting $\mathcal{H}$ follows the same proportions in the summary as they do in the original video. To do so, we propose the SumBal metric, which is a modified version of the Balance fairness metric generally employed in fair unsupervised learning tasks (Bera et al., 2019; Chierichetti et al., 2017):

$$\text{SumBal}(S, X, \mathcal{H}) = \min_{H^g \in \mathcal{H}} \min \left\{ R(S, X, H^g), \frac{1}{R(S, X, H^g)} \right\}$$

$$\text{where } R(S, X, H^g) = \sum_{i=1}^{n} \frac{H_i^g}{n} \Bigg/ \sum_{x_j \in S} \frac{H_j^g}{k} \tag{2}$$

Here, $R(S, X, H^g)$ is the ratio of the proportion of appearances of group/individual $g$ in the overall video to the generated summary $S$. We take the minimum between $R(S, X, H^g)$ and $1/R(S, X, H^g)$ to account for both under-representation and over-representation cases. Finally, SumBal returns the minimum over all groups/individuals and hence SumBal $\in [0, 1]$. We can take a simple example where we have a video with two individuals, A and B. Person A appears in 20% of the video frames and Person B appears in 50% of the frames, with 30% frames having no individuals. Now, assume that we generate a summary using a model which has the following proportions– Person A appears in 40% of the summary frames (over-representation)

---

[3]For brevity, at times we use the term protected groups to also refer to the set of individuals, but make it clear from context.

and Person B appears in 30% of the summary frames (under-representation). Then the SumBal term for Person A would be calculated as: $\min\{0.2/0.4, 0.4/0.2\} = 0.5$ and for Person B would be calculated as $\min\{0.5/0.3, 0.3/0.5\} = 0.6$. Since we take the minimum over all groups/individuals to calculate SumBal for the video, we get $\min\{0.5, 0.6\} = 0.5$ and the violating individual (with lowest fairness) is Person A.

### 3.3 Relating Fair Video Summarization to Fair Clustering

Given a dataset where each sample belongs to some protected group, the unsupervised task of group-level *fair clustering* involves partitioning samples in the dataset into *k clusters* according to some utility objective, while ensuring that each cluster has the same proportion of samples from each protected group as in the original dataset (Chhabra et al., 2021a; Chierichetti et al., 2017; Chhabra & Mohapatra, 2022; Bera et al., 2019). This is thematically similar to our notion of fairness in video summarization– video frames selected in the summary should have high utility, while ensuring that each protected group is represented in the summary output in the same proportions as in the original video.

There are also significant differences between these two problems: for instance, in fair clustering the entire dataset is selected, and the number of selections $k$ correspond to $k$ clusters where proportional fairness needs to be ensured for each cluster. However, for fair video summarization, part of the set of samples (that is, frames) are absent in the output, and $k$ corresponds to a subset of the dataset itself over which proportional fairness constraints need to be ensured. Moreover, in fair clustering, each sample is assumed to have some protected group membership, whereas video frames might have zero protected group appearances (for e.g. in the case when no individuals are present in a frame) changing the landscape of the problem considerably. Another issue stems from the value of $k$ itself– in clustering, $k$ is generally not very large whereas in video summarization $k$ can be order of magnitudes larger as it is typically set to be 15% of all video frames. It is well-known that for such large $k$, there is often clustering breakdown in cluster quality and computational efficiency (Fränti & Sieranoja, 2019; Pelleg et al., 2000). These challenges make it non-trivial to utilize fair clustering for fair video summarization directly without considerable modifications. However, clustering based approaches are often employed for data summarization (Ahmed, 2019), and we believe future work can exploit these connections to propose improved methods for fair video summarization as well.

## 4 Benchmarking Fairness Using *FairVidSum*

### 4.1 Curating Videos and Annotation Details

**Collecting Videos.** Our goal is to select videos that feature multiple individuals in diverse settings that allow us to annotate and account for fairness information. Moreover, we wanted *FairVidSum* to be similar to *TVSum* and *SumMe* so that existing video summarization models can utilize it in a plug-and-play manner. Similar to *TVSum* (Song et al., 2015), we collect videos from YouTube (Covington et al., 2016). We use the search terms "panel discussions", "podcasts", "interviews", "debates", "news", "discussions", and combinations of these keywords. We utilized these categories mainly because our primary requirement was to collect videos with human subjects from diverse backgrounds, which these categories guaranteed. Moreover, we wanted a certain number of individuals (at least $\geq 3$) in each video, which

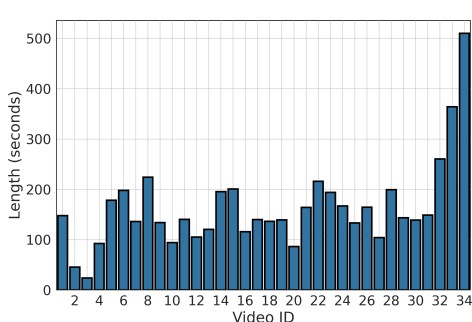

Figure 2: Video length distribution.

is also generally the case for these categories. Moreover, we restrict our videos to ones with a Creative Commons license, that lie between 1-4 minutes, and those that contain more than a single shot. Using this strategy we obtain 22 videos. While the *SumMe* dataset (Gygli et al., 2014) has no videos that meet this criteria, *TVSum* has a set of few videos (such as in the "documentaries" category) that we can use. In this manner, we also add another 12 videos from *TVSum* to *FairVidSum* and annotate them for fairness information. Thus, *FairVidSum* currently has a total of 34 videos, in line with current video summarization

datasets (SumMe has 25 videos and TVSum has 50 videos). We show the length distribution of these videos in Figure 2.

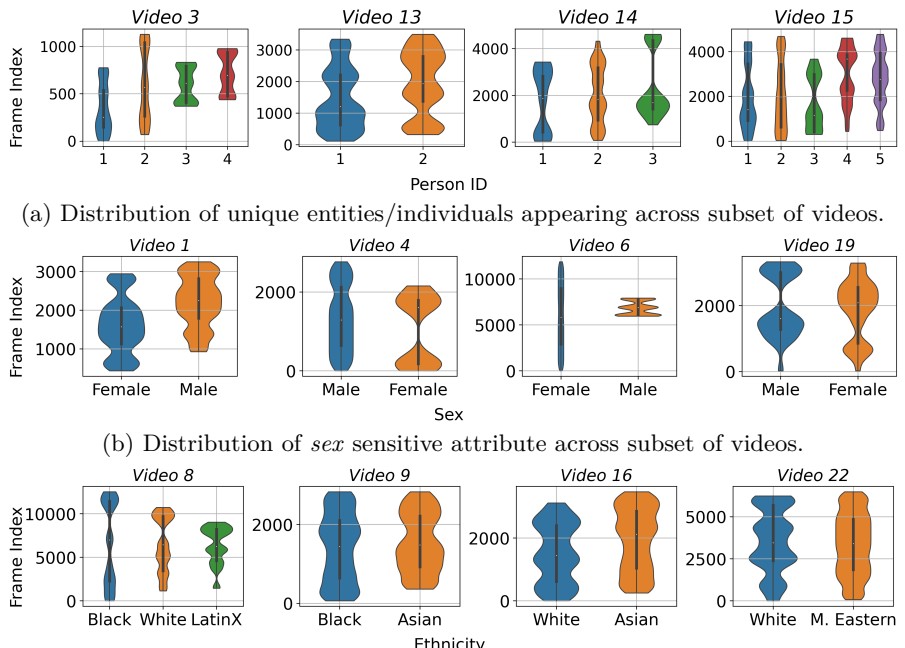

(a) Distribution of unique entities/individuals appearing across subset of videos.

(b) Distribution of *sex* sensitive attribute across subset of videos.

(c) Distribution of *ethnicity* sensitive attributes across subset of videos.

Figure 3: Violin plots showcasing the distribution of individuals and protected groups / sensitive attributes across randomly sampled videos.

**Annotating Videos with Importance Scores.** We follow much of the same procedure as used in *TVSum* (Song et al., 2015). We employ 10 annotators who consist of individuals from diverse fields in either graduate or post-graduate study. Annotators are first required to watch videos on mute in a single setting to ensure that the annotation scores are only based on visual information (Song et al., 2015). Similarly, to alleviate chronological bias (Song et al., 2015), frames are shuffled randomly. Next, to obtain scores annotators are shown uniformly sampled frames at 1/2 frames per second. Each annotator annotates every video and is required to label the provided frames with a score between 1 (least important) to 5 (most important) to be included in the summary. This task excludes the 12 *TVSum* additions as those already possess annotation scores. In this manner, we obtain 15400 responses total over all videos. Note that the number of annotators employed for this purpose is also satisfactory, as our annotation consistency analysis will later show.

**Annotating Videos with Fairness Information.** Other than these subjective annotations for importance scores, part of our dataset requires objective annotations for individuals and their sensitive attribute information. To do so we employ 4 annotators who collectively annotate all 34 videos with this information. Note that compared to annotation scores which are generally obtained for a subset of frames, fairness information needs to be collected for the entire video to calculate unfairness (such as using the SumBal metric). Thus, here, we annotate over 168120 frames total with information regarding different individuals appearing in frames and their sensitive attributes with respect to sex and ethnicity. For *sex* we annotate as *Male/Female* and for *ethnicity* we annotate for *White, Black, Middle Eastern, Asian,* and *Hispanic*.

## 4.2 Distribution of Individuals and Sensitive Attributes Across Videos

We aim to analyze the distribution of individuals appearing across videos. For this purpose, we randomly sample 8 videos out of 34, and plot the distributions of individuals as well as the distributions of *sex* and *ethnicity* protected groups in those videos as a function of their video frames using violin plots. These are visualized in Figure 3. The distributions for all the remaining videos are shown in Appendix A due to space

constraints. It is evident that both the number as well as frame-level distribution of individuals and protected groups varies widely across videos. This demonstrates one of the challenges associated with developing fair summarization models, as they need to be able to account for fairness in many diverse settings.

We also analyze group-level information for each video as a function of the annotation trend. Here, we can visualize the mean importance score for a video as a function of the frame indices, while also denoting sensitive attribute information for the frames. We demonstrate this for *Video 19* and *sex* as the protected group in Figure 4a and for *Video 16* with *ethnicity* as the protected group in Figure 4b. It can be seen that group-level information varies widely, and there is little correlation between importance scores and group-level information that would allow existing models to be fair.

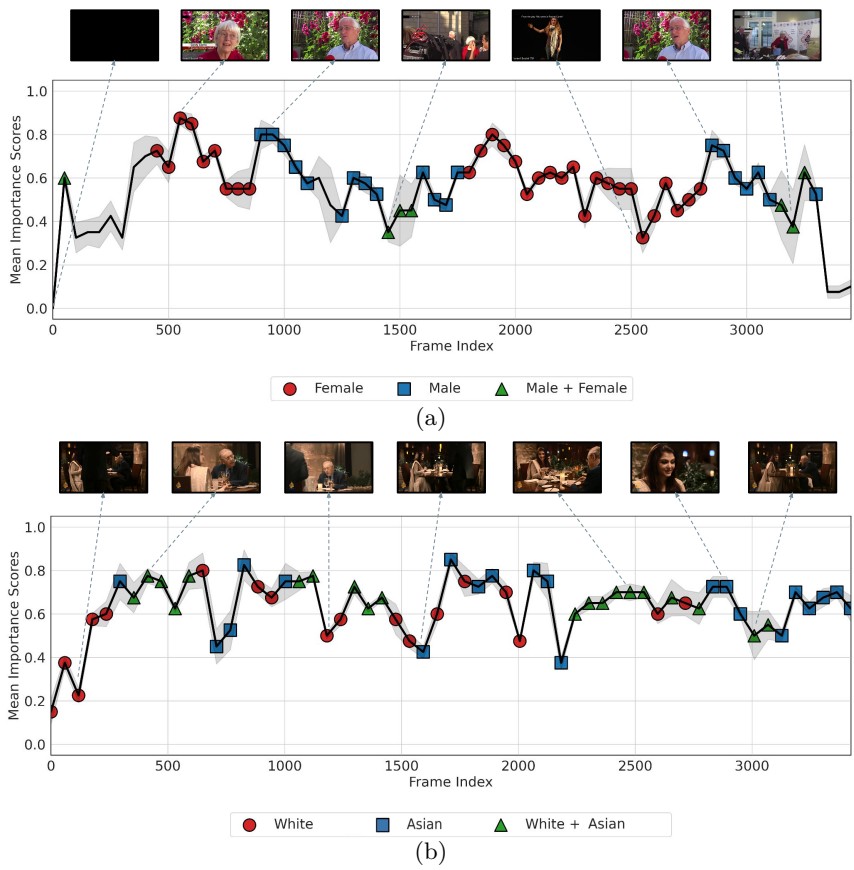

Figure 4: Mean importance (annotated) scores for (a) *Video 19* with protected group labels for *sex* and (b) for *Video 16* with protected group labels for *ethnicity*.

### 4.3 Annotator Consistency

We now cover another aspect of our dataset– the annotations, and their consistency. Annotator consistency with respect to video summarization is usually measured using the Cronbach's alpha (CA) (Cronbach, 1951). A higher CA value indicates more consistency among annotations. For *FairVidSum*, the CA value is 0.995. This is much higher than both *SumMe* (CA=0.74) and *TVSum* (CA=0.81). We posit that this is because 1) FairVidSum has a lot of intra-video homogeneity (a byproduct of our initial search terms) compared to both TVSum and SumMe that have more diversity in video categories, and 2) the visual complexity of certain TVSum/SumMe videos over FairVidSum results in more variance in annotation (for e.g. TVSum has some challenging videos in categories such as Dog Show and Flash Mob Gathering with rapid scene changes).

Annotator consistency can also be observed qualitatively for a given video. We can visualize this as a heatmap with rows as individual annotators, columns as respective video frames, and each cell thus representing the

annotator's importance score for that frame. We show this in Figure 5 for *Video 19*. It can be seen that for most frames, annotators agree on similar importance scores.

### 4.4 Discussion on Limitations

It is important to note that a number of possible improvements can be made to *FairVidSum* by analyzing and alleviating some of its current limitations. For instance, a current limitation of the dataset is the set of categories that the videos are sourced from ("panel discussions", "interviews", "debates", "news", among others). Videos from these categories often have repetitive frames and at least a few consistent individuals appear throughout the video. While these restricted source domain videos are useful for studying the fair video summarization problem as a first step, there are many other video categories that also might require fairness enforcement, such as those with a large number of individuals ($> 1000$) appearing in them for a very short duration of time (such as concert recordings, comedy shows, or large lectures). The challenges with enforcing fairness in such a setting are manifold: annotating fairness information due to the large number of individuals would

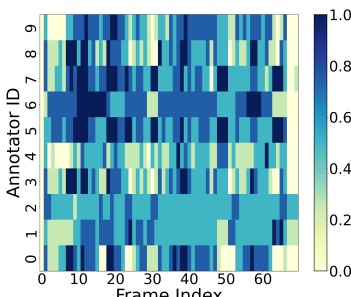

Figure 5: Annotator consistency matrix for *Video 19*.

be a non-trivial task and the large growth in $g$ (the number of groups/individuals) would possibly lead to small overall proportions for each group/individual resulting in marginal SumBal scores. For the latter, since SumBal scores might tend to zero with such large scale, new fairness evaluation metrics would need to be proposed as well. Moreover, incorporating videos from other source categories, such as those with rapid scene changes (for example sports sequences or movie montages), might lead to other unforseen challenges as well, both in data curation/annotation and in fairness enforcement. We defer the study and analysis of such problems to future work.

## 5 The FVS-LP Fair-Only Baseline

### 5.1 FVS-LP

We now present a baseline for fair video summarization– the Fair Video Summarization Linear Program (FVS-LP) which is a simple linear program (Schrijver, 1998) approach that only optimizes for fairness and selects frames such that the group proportions in the selected summary are as close as possible to the group proportions of the overall video. Note that this contribution is analogous to having a constant predictor in fair classification (Mehrabi et al., 2021)– as it predicts a constant it will always achieve maximum fairness, but low utility/accuracy. However, in fair video summarization, even such a simple fair-only baseline is not as conceptually straightforward as a constant/random predictor, and hence, we propose FVS-LP to showcase the fairness-utility tradeoff gap.

Let $\mathbf{0}_m$ and $\mathbf{1}_m$ denote an $m$ length vector of all zeros and all ones, respectively. We have a given video $V$ and its set of frames $X$, along with the set of group memberships $\mathcal{H}$. First, we transform $\mathcal{H}$ to matrix form for formulating the LP. Let $\mathcal{G} \in \{0,1\}^{n \times g}$ be derived from $\mathcal{H}$ such that each row vector $\mathcal{G}_i \in \{0,1\}^g$, $i \in [n]$ represents a frame and each of its entries are either 0 for absence or 1 for presence of a group in the frame. Let $0 \le \mathbf{x} \le 1$ be the optimization variable where each entry of $\mathbf{x} \in \mathbb{R}^n$ indicates if a frame is selected in the summary, then the LP can be written as Equation 3:

$$
\begin{aligned}
\text{minimize} \quad & \mathbf{0}_n^\top \mathbf{x} \\
\text{s.t.} \quad & \mathcal{G}^\top \mathbf{x} = k \cdot \frac{1}{n} \sum_{i=1}^n \mathcal{G}_i \\
& \mathbf{1}_n^\top \mathbf{x} = k \\
& 0 \le \mathbf{x} \le 1.
\end{aligned}
\tag{3}
$$

Note that since we are only optimizing for fairness, we do not care about utility and our optimization objective can simply be a vector of all zeros. Now, as is evident, the first constraint simply ensures that the sum of the

Table 1: Comparison of SOTA video summarization approaches on *FairVidSum*. The utility and fairness averages are calculated across all five splits. The violating groups that achieve the minimum fairness SumBal scores are also presented. Results on FVS-LP (our fairness baseline) along with Random and Human baselines are also provided. Blue/red indicates highest/lowest performance.

| Model | Type | Average $F_1$ Measure | SumBal (*Sex*) | | | SumBal (*Ethnicity*) | | | SumBal (*Individual*) | | |
|-------|------|-----------------------|---------|--------|-----------|---------|--------|-----------|---------|--------|-----------|
| | | | Average | Min | Violating | Average | Min | Violating | Average | Min | Violating |
| Random | - | 14.92 | 0.9497 | 0.8814 | *Male* (Vid. #30) | 0.9468 | 0.6670 | *Asian* (Vid. #33) | 0.8747 | 0.6670 | *Person 13* (Vid. #33) |
| Human | - | 68.91 | 0.4605 | 0.0000 | *Female* (Vid. #30) | 0.5503 | 0.0000 | *Asian* (Vid. #25) | 0.2773 | 0.0000 | *Person 2* (Vid. #18) |
| CA-SUM | Unsupervised | 62.78 | 0.5201 | 0.0000 | *Female* (Vid. #30) | 0.5468 | 0.0000 | *Asian* (Vid. #24) | 0.2441 | 0.0000 | *Person 4* (Vid. #8) |
| AC-SUM-GAN | Unsupervised | 64.33 | 0.5176 | 0.0000 | *Female* (Vid. #30) | 0.5455 | 0.0000 | *Asian* (Vid. #24) | 0.2616 | 0.0000 | *Person 4* (Vid. #8) |
| SUM-GAN-AAE | Unsupervised | 63.81 | 0.5222 | 0.1302 | *Female* (Vid. #26) | 0.5665 | 0.0000 | *Asian* (Vid. #24) | 0.2739 | 0.0000 | *Person 4* (Vid. #8) |
| SUM-GAN-SL | Unsupervised | 64.92 | 0.5254 | 0.0000 | *Female* (Vid. #30) | 0.5661 | 0.0000 | *Asian* (Vid. #24) | 0.2550 | 0.0000 | *Person 4* (Vid. #8) |
| SUM-IND | Unsupervised | 50.57 | 0.5677 | 0.0000 | *Female* (Vid. #24) | 0.5889 | 0.0000 | *Asian* (Vid. #24) | 0.2541 | 0.0000 | *Person 4* (Vid. #8) |
| DSNet | Supervised | 63.69 | 0.5358 | 0.0000 | *Female* (Vid. #30) | 0.5478 | 0.0000 | *Asian* (Vid. #24) | 0.2706 | 0.0000 | *Person 1* (Vid. #25) |
| VASNet | Supervised | 64.11 | 0.4622 | 0.0000 | *Female* (Vid. #25) | 0.5391 | 0.0000 | *Asian* (Vid. #24) | 0.2515 | 0.0000 | *Person 4* (Vid. #8) |
| PGL-SUM | Supervised | 63.75 | 0.4804 | 0.1042 | *Female* (Vid. #34) | 0.5374 | 0.0000 | *Asian* (Vid. #24) | 0.2575 | 0.0000 | *Person 4* (Vid. #8) |
| FVS-LP (*Sex*) | Unsupervised | 15.69 | 0.9987 | 0.9960 | *Female* (Vid. #4) | 0.7411 | 0.0000 | *Hispanic* (Vid.#8) | 0.3062 | 0.0000 | *Person 3* (Vid. #8) |
| FVS-LP (*Ethnicity*) | Unsupervised | 13.46 | 0.6642 | 0.0000 | *Female* (Vid. #25) | 0.9980 | 0.9822 | *Asian* (Vid. #33) | 0.2727 | 0.0000 | *Person 6* (Vid. #25) |
| FVS-LP (*Individual*) | Unsupervised | 14.13 | 0.9556 | 0.6289 | *Male* (Vid. #28) | 0.9471 | 0.6559 | *White* (Vid. #19) | 0.9932 | 0.9704 | *Person 6* (Vid. #25) |

selected samples' group memberships is equal to $k$ times the group proportions for the overall video. The second constraint ensures that the number of selected samples must be exactly $k$. After solving the above LP for $\mathbf{x}$, we can obtain the indices of summary frames selected from the set of frames $X$ by rounding the solution, as $I = \{i : \text{round}(\mathbf{x}_i) = 1\}$. Then, we can get the summary $S$ of video $V$ as $S = \{X_i : \forall i \in I\}$.

### 5.2 Mixing Sampling Strategy for Controlling Fairness-Utility Tradeoff

Since FVS-LP can be utilized to obtain fair summaries and current SOTA video summarization models can output summaries that have high utility, we propose a simple sampling strategy that *mixes* frames together from both of these to control the fairness-accuracy tradeoff.

We start with two distinct summaries: $S_{\text{acc}}$, generated from an existing model that optimizes for accuracy, and $S_{\text{fair}}$, generated using the FVS-LP baseline for fairness. To produce a summary that harmoniously blends both these objectives, we introduce the mixing strategy as follows. A mixing ratio, denoted as $\lambda$, determines the proportion of frames from the fairness-optimized summary $S_{\text{fair}}$ that will be integrated into the accuracy-optimized summary $S_{\text{acc}}$. Specifically, for a given $\lambda$, we randomly select $\lambda \cdot |S_{\text{acc}}|$ frames from $S_{\text{fair}}$. These selected frames are then incorporated into $S_{\text{acc}}$ by randomly substituting an equal number of included frames. Frames can then be sorted using timestamps. This procedure also ensures that the merged summary maintains the original summary length. Through this mixing sampling strategy, the resultant summary strikes a balance between the characteristics of accuracy and fairness.

## 6 Results

We now present results for benchmarking SOTA supervised and unsupervised models on *FairVidSum*. We utilize the following unsupervised models: CA-SUM (Apostolidis et al., 2022), AC-SUM-GAN (Apostolidis et al., 2020a), SUM-GAN-AAE (Apostolidis et al., 2020b), SUM-GAN-SL (Apostolidis et al., 2019), SUM-IND (Yaliniz & Ikizler-Cinbis, 2021) and the following supervised models: DSNet (Zhu et al., 2020), VASNet (Fajtl et al., 2019), PGL-SUM (Apostolidis et al., 2021b). Moreover, we also provide baseline results for a randomly generated summary (Random) and a summary generated using the knapsack algorithm on the average human annotated importance scores (Human). Finally, we also present results for FVS-LP while optimizing for each protected group type (*individual*, *sex*, and *ethnicity*). For each model/baseline, we provide the group members that achieve the minimum fairness values as well. Code, reproducibility, and miscellaneous dataset details are provided in Appendix D.

**Training and Evaluation.** We follow the standard evaluation procedure in existing video summarization literature, which involves randomly splitting the entire dataset into multiple parts or splits, typically 5,

Table 2: Comparison of SOTA video summarization model on *FairVidSum* for evaluation Split #1.

| Model | Type | Average $F_1$ Measure | SumBal (*Sex*) | | | SumBal (*Ethnicity*) | | | SumBal (*Individual*) | | |
|---|---|---|---|---|---|---|---|---|---|---|---|
| | | | Average | Min | Violating | Average | Min | Violating | Average | Min | Violating |
| Random | - | 14.74 | 0.9473 | 0.8818 | *Male* (Vid. #26) | 0.97 | 0.9415 | *White* (Vid. #26) | 0.8639 | 0.7477 | *Person 2* (Vid. #19) |
| Human | - | 67.08 | 0.4800 | 0.0000 | *Female* (Vid. #25) | 0.6717 | 0.0000 | *Asian* (Vid. #25) | 0.3080 | 0.0000 | *Person 1* (Vid. #25) |
| CA-SUM | Unsupervised | 62.11 | 0.5104 | 0.1044 | *Female* (Vid. #34) | 0.6746 | 0.4292 | *Asian* (Vid. #9) | 0.3042 | 0.0000 | *Person 1* (Vid. #25) |
| AC-SUM-GAN | Unsupervised | 63.99 | 0.4481 | 0.0000 | *Female* (Vid. #25) | 0.6467 | 0.0000 | *Asian* (Vid. #25) | 0.3261 | 0.0000 | *Person 1* (Vid. #25) |
| SUM-GAN-AAE | Unsupervised | 63.44 | 0.4887 | 0.1302 | *Female* (Vid. #26) | 0.6921 | 0.4567 | *Asian* (Vid. #9) | 0.2952 | 0.0000 | *Person 1* (Vid. #25) |
| SUM-GAN-SL | Unsupervised | 64.77 | 0.5298 | 0.2156 | *Male* (Vid. #26) | 0.7056 | 0.4567 | *Asian* (Vid. #9) | 0.2792 | 0.0000 | *Person 1* (Vid. #25) |
| SUM-IND | Unsupervised | 49.47 | 0.5415 | 0.3868 | *Male* (Vid. #26) | 0.6641 | 0.4503 | *Asian* (Vid. #9) | 0.3019 | 0.0000 | *Person 1* (Vid. #25) |
| DSNet | Supervised | 63.24 | 0.4432 | 0.1049 | *Female* (Vid. #31) | **0.6095** | 0.2742 | *Asian* (Vid. #25) | 0.3040 | 0.0000 | *Person 1* (Vid. #25) |
| VASNet | Supervised | **66.14** | 0.3386 | **0.0000** | *Female* (Vid. #25) | 0.5866 | **0.0000** | *Asian* (Vid. #25) | 0.2800 | 0.0000 | *Person 1* (Vid. #25) |
| PGL-SUM | Supervised | 65.18 | **0.4392** | 0.1042 | *Female* (Vid. #34) | 0.5869 | 0.2631 | *Asian* (Vid. #34) | **0.2701** | **0.0000** | *Person 1* (Vid. #25) |
| FVS-LP (*Sex*) | Unsupervised | 17.03 | **0.9990** | **0.9975** | *Female* (Vid. #25) | 0.8357 | 0.6559 | *White* (Vid. #19) | 0.4417 | 0.0000 | *Person 3* (Vid. #19) |
| FVS-LP (*Ethnicity*) | Unsupervised | **12.13** | 0.3967 | 0.0000 | *Female* (Vid. #25) | **0.9993** | **0.9983** | *Asian* (Vid. #25) | 0.3186 | 0.0000 | *Person 3* (Vid. #19) |
| FVS-LP (*Individual*) | Unsupervised | 16.81 | 0.9562 | 0.8250 | *Male* (Vid. #34) | 0.8999 | 0.6559 | *White* (Vid. #19) | **0.9930** | **0.9704** | *Person 6* (Vid. #25) |

each split subjected to an 80:20 train/test partitioning (Apostolidis et al., 2022; 2020a;b; Zhu et al., 2020; Fajtl et al., 2019; Apostolidis et al., 2021b; Kanafani et al., 2021). The models are trained on the training set of a given split and subsequently evaluated on the corresponding test set within the same split. A detailed breakdown of the video distribution for all 5 train/test splits are present in the Appendix B. The $F_1$-measure evaluates the similarity between a model predicted summary and a user-defined summary by assessing their overlap. The $F_1$ scores are calculated for each individual video and then averaged over the entirety of a given split which are then averaged across all 5 splits. SumBal is also evaluated per video, and the same procedure is followed to obtain averages.

**Details on Model Training.** We downsample videos to 1/2 frames per second as our video frames are often repetitive. Following prior work, we utilize GoogleNet (Szegedy et al., 2015) (trained on ImageNet) to extract frame features from the *pool5* layer, which outputs a dimensionality of 1024. When training the various models, we adhere to their original procedures, and generally employ default settings and hyperparameters. Any alterations or adjustments to the default training parameters are detailed in Appendix B. To ensure a fair comparison, we employ the same splits for training/testing across all models.

*FairVidSum* **Benchmarking Results.** Since we have 5 evaluation splits, we present average results over all splits in Table 1. We also present results for Split #1 in Table 2 and for the remaining evaluation splits in Appendix C due to space constraints. As can be seen in Tables 1 and 2 both unsupervised and supervised SOTA models tend to achieve high utility performance computed in terms of the $F_1$-measure ($> 60$) averaged over all videos and all splits.[4] However, these models have low fairness performance, with minimum SumBal scores for all three group types: *sex*, *ethnicity*, and *individuals* most often tend to be 0 and generally $< 0.5$. Interestingly, the group members that achieve the lowest fairness values across all splits and videos tend to consistently be *Female* for *sex* as the protected group, *Asian* for *ethnicity* as the protected group, and *Person 4* for *individual* fairness. We also present average SumBal scores which are higher at times, but have very large variance showcasing that models are not inherently optimizing for fairness. The human annotated summary also fares similarly to the SOTA models, as it is only annotated for performance. Moreover, the randomly generated summary has very low utility performance scores– typically with $F_1$-measure values less than 15 which follows the fact that summary frames are picked completely at random. However, the random summary has high fairness scores. We hypothesize that this is the case because by picking frames uniformly at random, the probability that each group member is picked according to their proportions is uniform in expectation. As a result, random frame selection leads to improved fairness.

Furthermore, we provide results for three versions of FVS-LP, each instantiated to optimize one type of protected group/sensitive attribute. For each of these, FVS-LP achieves the highest fairness performance across all models and baselines for the group it is optimizing for. However, it does not lead to good utility performance, which is to be expected as it is only directly optimizing for fairness. This implies that while

---

[4]This utility performance is in line with SOTA results for *TVSum* and *SumMe*; refer to (Song et al., 2015; Apostolidis et al., 2021a) for details.

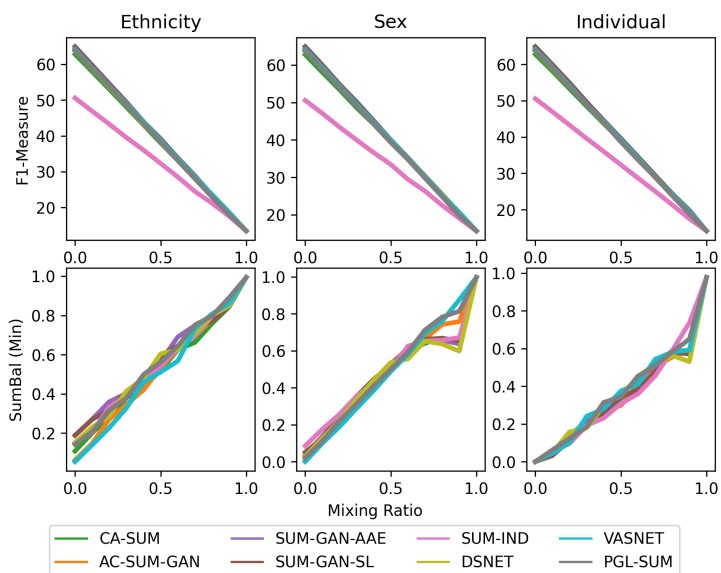

Figure 6: Results for the mixing sampling strategy averaged over all splits. The top row demonstrates the effect of increasing mixing ratio ($\lambda$) on utility ($F_1$ Measure) whereas the bottom row showcases the effect on fairness (SumBal). Each column represents mixing done for a particular protected group (ethnicity/sex/individual). $\lambda$ is varied in increments of 0.1 from 0 to 1.

there is a gap in fairness that can be optimized for, optimizing for both fairness and performance is a non-trivial task. For future work, methods that jointly optimize both fairness and utility can thus be proposed. Note that the trends between the average performance and Split #1 are very similar, and this is also the case for the other evaluation splits (Appendix C). Generally, we observe that supervised models tend to exhibit lower average SumBal values. This trend might be a direct consequence of these models' learning process, which strives to closely align with human or ground truth summaries that, as previously mentioned, are solely optimized for utility. This observation further underscores the importance of incorporating a fairness evaluation and learning criterion in the model design and training process. Another crucial insight from our benchmarking analysis is the distinct difficulty in upholding *individual* fairness. This is clearly evident by the consistently lowest average SumBal values (compared with *sex* and *ethnicity*) and predominant minimum values of zero. A SumBal value of zero essentially indicates that a group or individual, though present in the original video, has been completely excluded from the generated summary.

**Mixing Strategy Results to Showcase the Fairness-Utility Tradeoff.** We now presents results for the mixing sampling strategy based on FVS-LP described in Section 5.2. Results averaged over all splits are shown in Figure 6 and for each of the individual splits in Appendix C.2. In Figure 6 each column indicates mixing undertaken for a particular protected group: ethnicity, sex, and individuals. Each SOTA model whose original utility summary the fair summary (FVS-LP for the particular protected group) is mixed with is shown as an individual line. The top row of the figure showcases utility results ($F_1$ Measure) and the bottom row denotes fairness (minimum value of SumBal obtained). It can be seen that as the mixing ratio $\lambda$ is increased from 0 to 1, in increments of 0.1, the utility starts to decrease and fairness increases. Thus, $\lambda$ can be used as a fairness-utility tradeoff for balancing fairness and utility. Results for the individual splits in Appendix C.2 exhibit similar trends.

**Discussion on Results.** Video summarization models are designed to extract the most representative and salient content from videos, emphasizing features like motion, distinct objects, and long-range temporal sequences. Current models use mechanisms such as self-attention (Vaswani et al., 2017), GANs (Goodfellow et al., 2020), and LSTMs (Hochreiter & Schmidhuber, 1997) to often drive this extraction process, seeking to replicate human-like summaries that focus on pivotal moments and high-saliencey content.

We believe that the sole focus on saliency results in the fairness issues exhibited by these models. Models may prioritize certain segments or individuals that align with their understanding of importance, which is

often derived from visual and temporal cues. For instance, high-contrast or motion-intensive sequences might overshadow static, less visually distinctive moments, regardless of the duration or frequency of appearance of entities within those moments. Furthermore, most, if not all, of the individual components (GANs, LSTMs, transformers, etc.) used by models have been shown to exhibit bias and unfairness in prior research (refer to (Kenfack et al., 2021; Sun et al., 2021; Qiang et al., 2023) for more details).

Moreover, human annotations, which serve as ground truths for models, might themselves be biased. Human annotators may not be focused on representative fairness, but only on capturing what they perceive as the most noteworthy moments, causing models to adopt similar biases. This is evidenced by the slightly lower average SumBal scores for fully supervised models which largely rely on ground truth annotations.

To summarize: fairness aims for a proportional representation of entities or sensitive attributes while utility focuses on capturing the most informative and representative moments guided by visual and temporal cues. Thus, we believe that models optimized for accuracy may inadvertently neglect frequently occurring yet less visually impactful segments and vice versa.

## 7 Conclusion

In this paper, we propose the fair video summarization problem which has thematic connections to the problem of fair clustering. We introduce the *FairVidSum* dataset, which consists of videos with annotations of frame-level importance scores, appearance of individuals across frames, as well as information regarding their sensitive attributes such as sex and ethnicity. We also propose the SumBal metric, which measures the disparity in fairness of the generated summary with regards to the original video. Through *FairVidSum* and SumBal, we benchmark a number of existing SOTA video summarization models and find that these generate highly unfair summaries as they do not directly optimize for fairness. Finally, we also propose the FVS-LP method which is a linear programming baseline optimized only for fairness, analogous to a constant predictor in fair classification.

Our paper constitutes the first work on fair video summarization, and hence, impacts the community in numerous ways[5]. It is important to ensure that learning algorithms account for equitable representation. Through this work, we seek to bridge this gap for the task of video summarization. Furthermore, there are multiple avenues for future work. Better fair video summarization models can be developed using *FairVidSum*, which improve upon both accuracy and fairness. Pre- and post-processing fair approaches can also be proposed. Methods that optimize for multiple protected groups jointly can also be investigated. Novel fairness definitions can also be proposed, similar to the trend observed in the fair clustering community (Chhabra et al., 2021a).

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

# Appendix

## A   Additional Violin Plots for Remaining Videos

We provide remaining violin plots for all videos in *FairVidSum* to visualize the distribution of unique individuals (Figure 7), *sex* sensitive attribute (Figure 8), and *ethnicity* sensitive attribute (Figure 9). These distributions highlight the importance and difficulty of summarizing videos while ensuring fairness. The individual plots, in particular, showcase the most challenging scenarios, as they contain videos (such as Vid. #24, #25, #30, #33) with numerous individuals appearing in very limited frames. Consequently, any missing individuals would result in an unfair summary and a SumBal score of zero. The plots emphasize the need to capture and represent proportionality and fairness in the video summaries.

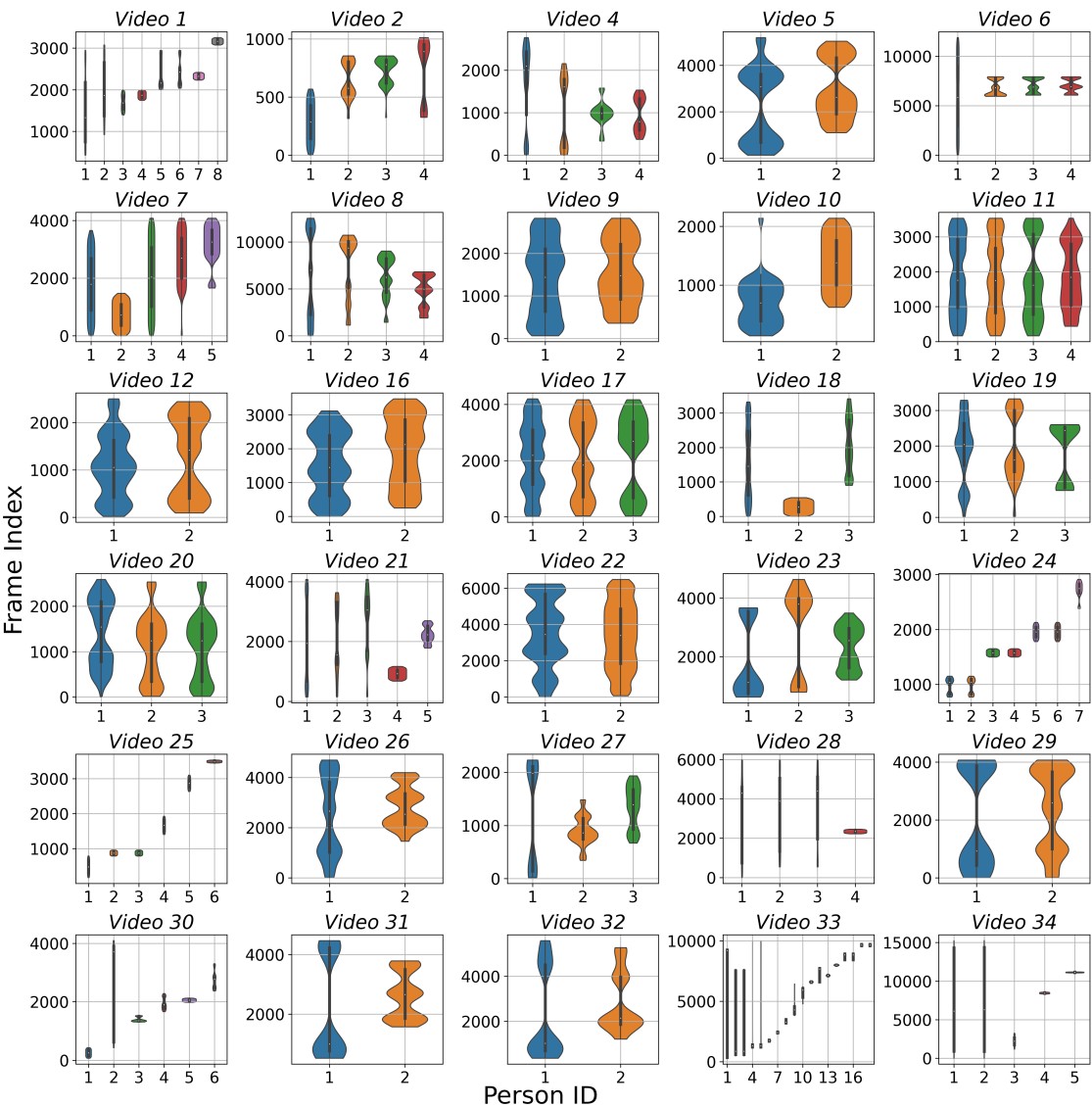

Figure 7: Violin plots showcasing the distribution of unique *entities/individuals* appearing across all videos. *Videos #3, #13, #14, #15* are present in main paper.

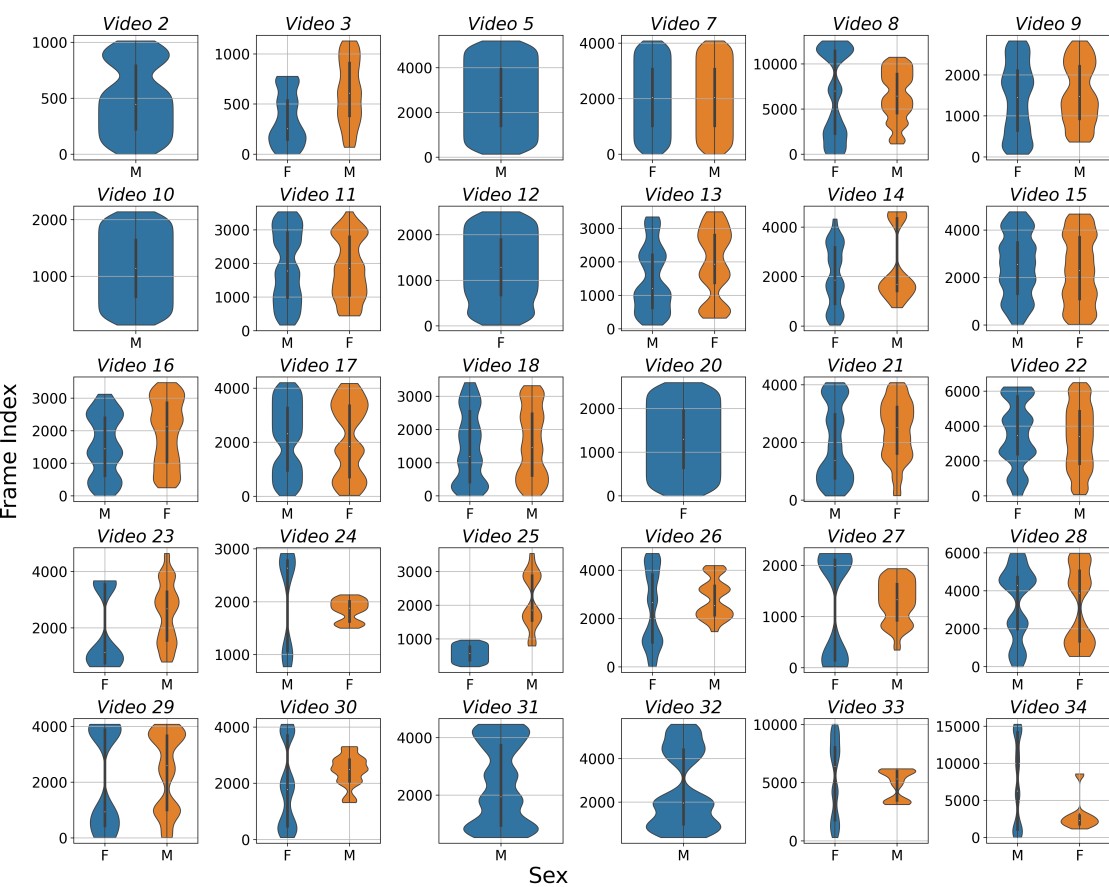

Figure 8: Violin plots showcasing the distribution *sex* sensitive attribute across all videos. Here M denotes *Male*, and F denotes *Female. Videos #1, #4, #6, #19* are present in main paper.

# B Additional Details Regarding Model Training and Evaluation

## B.1 Evaluation Splits

In accordance with previous research methodologies, we have created five splits, chosen randomly. These splits follow an 80:20 partitioning for training and testing, respectively. With 34 videos total, each split comprises 28 videos in the training set and 6 in the testing set. We provide the test sets for every split used in our benchmarks below. It is important to note that any videos not included in the test sets for each split would naturally be part of the corresponding training sets. The splits are as follows: *Split #1 Test set*: Vid. #9, Vid. #11, Vid. #19, Vid. #25, Vid. #26, Vid. #34; *Split #2 Test set*: Vid. #2, Vid. #8, Vid. #20, Vid. #24, Vid. #29, Vid. #32; *Split #3 Test set*: Vid. #8, Vid. #18, Vid. #20, Vid. #24, Vid. #28, Vid. #33; *Split #4 Test set*: Vid. #4, Vid. #8, Vid. #17, Vid. #18, Vid. #29, Vid. #30; *Split #5 Test set*: Vid. #3, Vid. #15, Vid. #16, Vid. #20, Vid. #25, Vid. #28.

## B.2 Models and Training Details

We follow the original training procedures established for all benchmarked models and adjustments to specific hyperparameters are described below. Any parameters not explicitly mentioned were maintained at the default values specified in the original code or paper for each model. For DSNet, we use the anchor-free model.

All adjustments made to parameters to train models on *FairVidSum*:

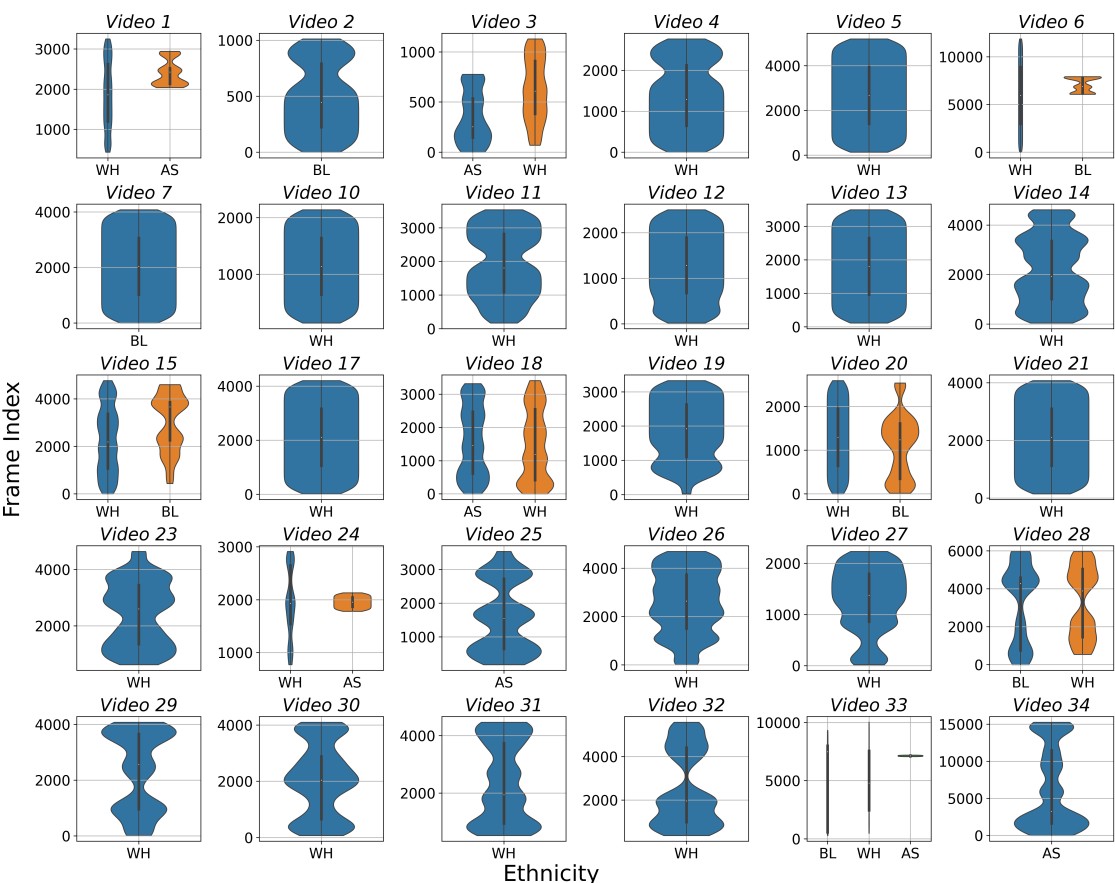

Figure 9: Violin plots showcasing the distribution *ethnicity* sensitive attribute across all videos. Here WH denotes *White*, BL denotes *Black*, and AS denotes *Asian*. *Videos #8, #9, #16, #22* are present in main paper.

- AC-SUM-GAN: `regularization_factor` = 5.0, `clip` = 1.0, `action_state_size` = 8

- CA-SUM: `block_size` = 60, `init_gain` = 1.0, `n_epochs` = 200, `clip` = 1.0, `lr` = 1e-4, `l2_req` = 1e-6, `reg_factor` = 5.0

- PGL-SUM: `clip` = 1.0, `lr` = 1e-4, `l2_req` = 1e-4

- SUM-GAN-AAE: `clip` = 1.0, `hidden_size` = 512, `regularization_factor` = 5.0, `lr` = 1e-5

- SUM-GAN-SL: `clip` = 1.0, `hidden_size` = 512, `regularization_factor` = 5.0

## C   Results on Remaining Evaluation Splits

### C.1   Benchmarking Results

Here, we present results for the remaining evaluation splits: Split #2, Split #3, Split #4, and Split #5 as Tables 3, 4, 5, and 6, respectively. The results show similar trends to the averages table and Split #1 table presented in the main text.

Table 3: Comparison of SOTA video summarization models on *FairVidSum* for evaluation Split #2.

| Model | Type | Average $F_1$ Measure | SumBal (*Sex*) | | | SumBal (*Ethnicity*) | | | SumBal (*Individual*) | | |
|---|---|---|---|---|---|---|---|---|---|---|---|
| | | | Average | Min | Violating | Average | Min | Violating | Average | Min | Violating |
| Random | - | 14.95 | 0.9837 | 0.9555 | *Female* (Vid. #29) | 0.9696 | 0.9199 | *Asian* (Vid. #24) | 0.8866 | 0.7417 | *Person 3* (Vid. #24) |
| Human | - | 65.00 | 0.4551 | 0.0000 | *Female* (Vid. #24) | 0.4768 | 0.0000 | *Asian* (Vid. #24) | 0.2909 | 0.0000 | *Person 1* (Vid. #24) |
| CA-SUM | Unsupervised | 60.77 | 0.5449 | 0.0000 | *Female* (Vid. #24) | 0.4971 | 0.0000 | *Asian* (Vid. #24) | 0.1819 | 0.0000 | *Person 4* (Vid. #8) |
| AC-SUM-GAN | Unsupervised | 62.41 | 0.5362 | 0.0000 | *Female* (Vid. #24) | 0.4573 | 0.0000 | *Asian* (Vid. #24) | 0.2634 | 0.0000 | *Person 3* (Vid. #24) |
| SUM-GAN-AAE | Unsupervised | **61.49** | 0.5213 | 0.0000 | *Female* (Vid. #24) | 0.4658 | 0.0000 | *Asian* (Vid. #24) | 0.1835 | 0.0000 | *Person 4* (Vid. #8) |
| SUM-GAN-SL | Unsupervised | 62.02 | 0.4958 | 0.0000 | *Female* (Vid. #24) | 0.4529 | 0.0000 | *Asian* (Vid. #24) | **0.1714** | **0.0000** | *Person 4* (Vid. #8) |
| SUM-IND | Unsupervised | 49.80 | 0.6805 | 0.0000 | *Female* (Vid. #24) | 0.6940 | 0.0000 | *Asian* (Vid. #24) | 0.3570 | 0.0000 | *Person 1* (Vid. #2) |
| DSNet | Supervised | 62.45 | 0.5661 | 0.0000 | *Female* (Vid. #24) | 0.5032 | 0.0000 | *Asian* (Vid. #24) | 0.2888 | 0.0000 | *Person 4* (Vid. #8) |
| VASNet | Supervised | 60.53 | 0.4796 | 0.0000 | *Female* (Vid. #24) | 0.4676 | 0.0000 | *Asian* (Vid. #24) | 0.1963 | 0.0000 | *Person 4* (Vid. #8) |
| PGL-SUM | Supervised | 61.04 | **0.4627** | **0.0000** | *Female* (Vid. #24) | **0.4300** | **0.0000** | *Asian* (Vid. #24) | 0.1881 | 0.0000 | *Person 4* (Vid. #8) |
| FVS-LP (*Sex*) | Unsupervised | 15.30 | **0.9985** | **0.9968** | *Male* (Vid. #24) | 0.7033 | 0.0000 | *Hispanic* (Vid. #8) | 0.4183 | 0.0000 | *Person 1* (Vid. #2) |
| FVS-LP (*Ethnicity*) | Unsupervised | 14.70 | 0.8599 | 0.4721 | *Female* (Vid. #29) | **0.9981** | **0.9934** | *Asian* (Vid. #24) | 0.3755 | 0.0000 | *Person 1* (Vid. #2) |
| FVS-LP (*Individual*) | Unsupervised | **12.41** | 0.9714 | 0.8350 | *Male* (Vid. #2) | 0.9648 | 0.8350 | *Black* (Vid. #2) | **0.9963** | **0.9905** | *Person 2* (Vid. #2) |

Table 4: Comparison of SOTA video summarization models on *FairVidSum* for evaluation Split #3.

| Model | Type | Average $F_1$ Measure | SumBal (*Sex*) | | | SumBal (*Ethnicity*) | | | SumBal (*Individual*) | | |
|---|---|---|---|---|---|---|---|---|---|---|---|
| | | | Average | Min | Violating | Average | Min | Violating | Average | Min | Violating |
| Random | - | 14.96 | 0.9477 | 0.8758 | *Female* (Vid. #24) | 0.8942 | 0.6670 | *Asian* (Vid. #33) | 0.8614 | 0.6670 | *Person 13* (Vid. #33) |
| Human | - | 67.61 | 0.4784 | 0.0000 | *Female* (Vid. #24) | 0.4518 | 0.0000 | *Asian* (Vid. #24) | 0.2097 | 0.0000 | *Person 2* (Vid. #18) |
| CA-SUM | Unsupervised | 59.76 | 0.5547 | 0.0000 | *Female* (Vid. #24) | **0.3859** | **0.0000** | *Asian* (Vid. #24) | 0.1901 | 0.0000 | *Person 4* (Vid. #8) |
| AC-SUM-GAN | Unsupervised | 61.68 | 0.6236 | 0.0000 | *Female* (Vid. #24) | 0.4123 | 0.0000 | *Asian* (Vid. #24) | 0.1999 | 0.0000 | *Person 4* (Vid. #8) |
| SUM-GAN-AAE | Unsupervised | 61.63 | 0.5859 | 0.0000 | *Female* (Vid. #24) | 0.4291 | 0.0000 | *Asian* (Vid. #24) | 0.2779 | 0.0000 | *Person 1* (Vid. #24) |
| SUM-GAN-SL | Unsupervised | **62.77** | 0.6211 | 0.0000 | *Female* (Vid. #24) | 0.4122 | 0.0000 | *Asian* (Vid. #24) | 0.2767 | 0.0000 | *Person 3* (Vid. #24) |
| SUM-IND | Unsupervised | 44.79 | **0.5061** | **0.0000** | *Female* (Vid. #24) | 0.4214 | 0.0000 | *Asian* (Vid. #24) | **0.1481** | **0.0000** | *Person 4* (Vid. #8) |
| DSNet | Supervised | 60.16 | 0.5448 | 0.0000 | *Female* (Vid. #26) | 0.4364 | 0.0000 | *Asian* (Vid. #24) | 0.2304 | 0.0000 | *Person 4* (Vid. #8) |
| VASNet | Supervised | 60.93 | 0.5646 | 0.0000 | *Female* (Vid. #24) | 0.4551 | 0.0000 | *Asian* (Vid. #24) | 0.2858 | 0.0000 | *Person 1* (Vid. #24) |
| PGL-SUM | Supervised | 59.15 | 0.5117 | 0.0000 | *Female* (Vid. #24) | 0.4086 | 0.0000 | *Asian* (Vid. #24) | 0.1965 | 0.0000 | *Person 4* (Vid. #8) |
| FVS-LP (*Sex*) | Unsupervised | 17.03 | **0.9990** | **0.9968** | *Male* (Vid. #24) | 0.5242 | 0.0000 | *Hispanic* (Vid. #8) | 0.0939 | 0.0000 | *Person 3* (Vid. #8) |
| FVS-LP (*Ethnicity*) | Unsupervised | 16.06 | 0.7394 | 0.0000 | *Male* (Vid. #33) | **0.9951** | **0.9822** | *Asian* (Vid. #33) | 0.1387 | 0.0000 | *Person 4* (Vid. #8) |
| FVS-LP (*Individual*) | Unsupervised | **16.03** | 0.9371 | 0.6289 | *Male* (Vid. #28) | 0.9373 | 0.6499 | *White* (Vid. #28) | **0.9923** | **0.9731** | *Person 14* (Vid. #33) |

## C.2 Mixing Strategy Results

Here we present results for the mixing sampling strategy described in Section 5.2 for each of the individual splits as Figures 10 (Split #1), 11 (Split #2), 12 (Split #3), 13 (Split #4), and 14 (Split #5). The mixing ratio $\lambda$ is varied from 0 to 1 in increments of 0.1.

# D Code, Reproducibility, and Miscellaneous Details

## D.1 Github Repository

The Github repository contains all the code needed for reproducing experiments, and also hosts the *FairVid-Sum* dataset. It is located here: `https://github.com/anshuman23/fair_video_summarization_tmlr`

## D.2 Environment Specifications

We use Python 3.8.16 and Anaconda to install all required libraries to run all models. The Anaconda environment yaml file is provided in our repository. The experiments were conducted on Ubuntu 20.04 using NVIDIA GeForce RTX 3070 GPUs (CUDA version 11.1).

## D.3 Code Details

The training codes utilized in our experiments were directly obtained from the official github repositories of the respective models, all implemented in PyTorch (v1.12.1). We condense codes into less files for ease

Table 5: Comparison of SOTA video summarization models on *FairVidSum* for evaluation Split #4.

| Model | Type | Average $F_1$ Measure | SumBal (*Sex*) Average | Min | Violating | SumBal (*Ethnicity*) Average | Min | Violating | SumBal (*Individual*) Average | Min | Violating |
|---|---|---|---|---|---|---|---|---|---|---|---|
| Random | - | 16.08 | 0.9324 | 0.8814 | *Male* (Vid. #30) | 0.9760 | 0.9043 | *White* (Vid. #18) | 0.8886 | 0.7253 | *Person 3* (Vid. #30) |
| Human | - | 76.51 | 0.3735 | 0.0000 | *Female* (Vid. #30) | 0.5895 | 0.0752 | *White* (Vid. #30) | 0.2522 | 0.0000 | *Person 2* (Vid. #18) |
| CA-SUM | Unsupervised | 71.89 | 0.4336 | 0.0000 | *Female* (Vid. #30) | **0.6222** | **0.0750** | *White* (Vid. #30) | 0.2501 | 0.0000 | *Person 4* (Vid. #8) |
| AC-SUM-GAN | Unsupervised | 74.02 | **0.4258** | **0.0000** | *Female* (Vid. #30) | 0.6222 | 0.2741 | *White* (Vid. #30) | **0.2026** | **0.0000** | *Person 4* (Vid. #8) |
| SUM-GAN-AAE | Unsupervised | 72.49 | 0.5096 | 0.0000 | *Female* (Vid. #30) | 0.6994 | 0.4450 | *White* (Vid. #29) | 0.2823 | 0.0000 | *Person 4* (Vid. #8) |
| SUM-GAN-SL | Unsupervised | **73.42** | 0.4458 | 0.0000 | *Female* (Vid. #30) | 0.6988 | 0.4450 | *White* (Vid. #29) | 0.2615 | 0.0000 | *Person 4* (Vid. #8) |
| SUM-IND | Unsupervised | 60.96 | 0.5692 | 0.0000 | *Male* (Vid. #29) | 0.6363 | 0.2419 | *White* (Vid. #29) | 0.2004 | 0.0000 | *Person 4* (Vid. #8) |
| DSNet | Supervised | 71.53 | 0.5743 | 0.0000 | *Female* (Vid. #30) | 0.6661 | 0.4450 | *White* (Vid. #29) | 0.2412 | 0.0000 | *Person 3* (Vid. #8) |
| VASNet | Supervised | 69.95 | 0.4444 | 0.0000 | *Female* (Vid. #30) | 0.6470 | 0.2719 | *White* (Vid. #30) | 0.2055 | 0.0000 | *Person 4* (Vid. #8) |
| PGL-SUM | Supervised | 70.03 | 0.5058 | 0.0000 | *Female* (Vid. #30) | 0.7343 | 0.4450 | *White* (Vid. #29) | 0.3027 | 0.0000 | *Person 4* (Vid. #8) |
| FVS-LP (*Sex*) | Unsupervised | 16.69 | **0.9981** | **0.9960** | *Female* (Vid. #4) | 0.8231 | 0.0000 | *Hispanic* (Vid. #8) | 0.3169 | 0.0000 | *Person 3* (Vid. #4) |
| FVS-LP (*Ethnicity*) | Unsupervised | **14.49** | 0.5619 | 0.0000 | *Female* (Vid. #4) | **0.9994** | **0.9987** | *White* (Vid. #18) | 0.2257 | 0.0000 | *Person 2* (Vid. #4) |
| FVS-LP (*Individual*) | Unsupervised | 15.90 | 0.9762 | 0.8695 | *Male* (Vid. #17) | 0.9926 | 0.9621 | *White* (Vid. #29) | **0.9949** | **0.9820** | *Person 5* (Vid. #30) |

Table 6: Comparison of SOTA video summarization models on *FairVidSum* for evaluation Split #5.

| Model | Type | Average $F_1$ Measure | SumBal (*Sex*) Average | Min | Violating | SumBal (*Ethnicity*) Average | Min | Violating | SumBal (*Individual*) Average | Min | Violating |
|---|---|---|---|---|---|---|---|---|---|---|---|
| Random | - | 13.96 | 0.9375 | 0.8692 | *Male* (Vid. #3) | 0.9242 | 0.8692 | *White* (Vid. #3) | 0.8732 | 0.7020 | *Person 2* (Vid. #25) |
| Human | - | 68.34 | 0.5157 | 0.0000 | *Female* (Vid. #25) | 0.5615 | 0.0000 | *Asian* (Vid. #25) | 0.3257 | 0.0000 | *Person 1* (Vid. #25) |
| CA-SUM | Unsupervised | 59.34 | 0.5567 | 0.0384 | *Female* (Vid. #3) | 0.5542 | 0.0384 | *Asian* (Vid. #3) | 0.2944 | 0.0000 | *Person 1* (Vid. #25) |
| AC-SUM-GAN | Unsupervised | 59.56 | 0.5545 | 0.0390 | *Female* (Vid. #3) | 0.5890 | 0.0390 | *Asian* (Vid. #3) | 0.3159 | 0.0000 | *Person 1* (Vid. #25) |
| SUM-GAN-AAE | Unsupervised | 60.01 | 0.5053 | 0.0401 | *Female* (Vid. #3) | 0.5460 | 0.0401 | *Asian* (Vid. #3) | 0.3304 | 0.0000 | *Person 1* (Vid. #25) |
| SUM-GAN-SL | Unsupervised | 61.59 | 0.5346 | 0.0384 | *Female* (Vid. #3) | 0.5613 | 0.0384 | *Asian* (Vid. #3) | 0.2863 | 0.0000 | *Person 1* (Vid. #25) |
| SUM-IND | Unsupervised | 47.81 | 0.5411 | 0.0390 | *Female* (Vid. #3) | 0.5287 | 0.0390 | *Asian* (Vid. #3) | **0.2629** | **0.0000** | *Person 1* (Vid. #25) |
| DSNet | Supervised | 61.06 | 0.5506 | 0.0401 | *Female* (Vid. #3) | **0.5238** | 0.0411 | *Asian* (Vid. #3) | 0.2883 | 0.0000 | *Person 1* (Vid. #25) |
| VASNet | Supervised | 62.99 | 0.4836 | 0.0000 | *Female* (Vid. #25) | 0.5393 | 0.0000 | *Asian* (Vid. #25) | 0.2897 | 0.0000 | *Person 1* (Vid. #25) |
| PGL-SUM | Supervised | **63.37** | **0.4828** | **0.0000** | *Female* (Vid. #25) | 0.5274 | **0.0000** | *Asian* (Vid. #25) | 0.3304 | 0.0000 | *Person 1* (Vid. #25) |
| FVS-LP (*Sex*) | Unsupervised | 12.42 | **0.9988** | **0.9975** | *Female* (Vid. #25) | 0.8192 | 0.5633 | *Black* (Vid. #20) | 0.2604 | 0.0000 | *Person 3* (Vid. #3) |
| FVS-LP (*Ethnicity*) | Unsupervised | 9.928 | 0.7632 | 0.0000 | *Female* (Vid. #25) | **0.9983** | **0.9974** | *Black* (Vid. #28) | 0.3052 | 0.0000 | *Person 3* (Vid. #3) |
| FVS-LP (*Individual*) | Unsupervised | **9.512** | 0.9367 | 0.6289 | *Male* (Vid. #28) | 0.9406 | 0.6499 | *White* (Vid. #28) | **0.9894** | **0.9704** | *Person 6* (Vid. #25) |

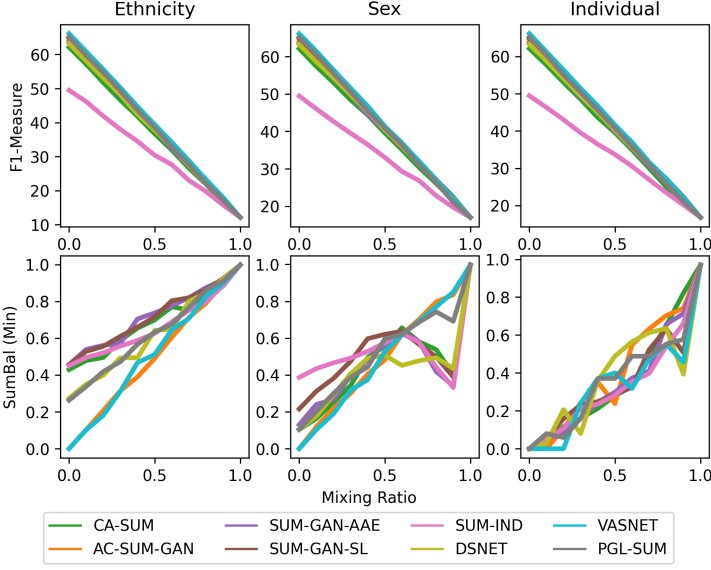

Figure 10: Results for the mixing sampling strategy for Split #1.

of running. We use our dataset splits (randomly generated) to evaluate and train all models (also provided in repository). We selected the trained models that achieved the highest $F_1$ scores per split (which was

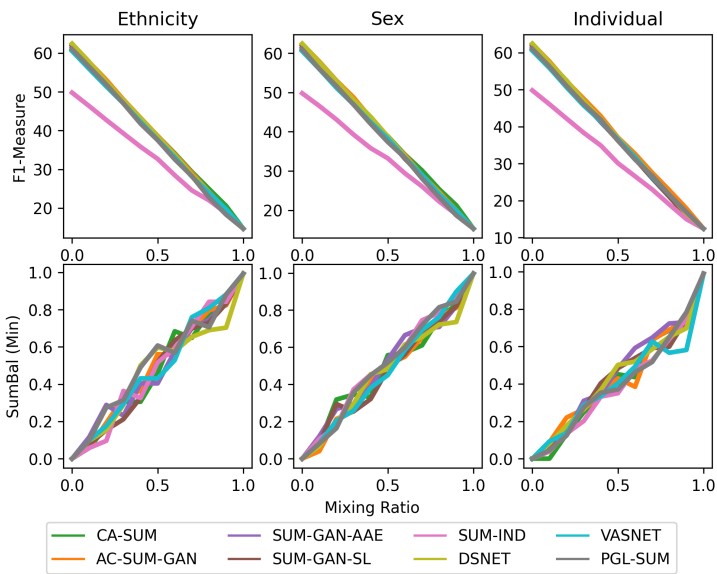

Figure 11: Results for the mixing sampling strategy for Split #2.

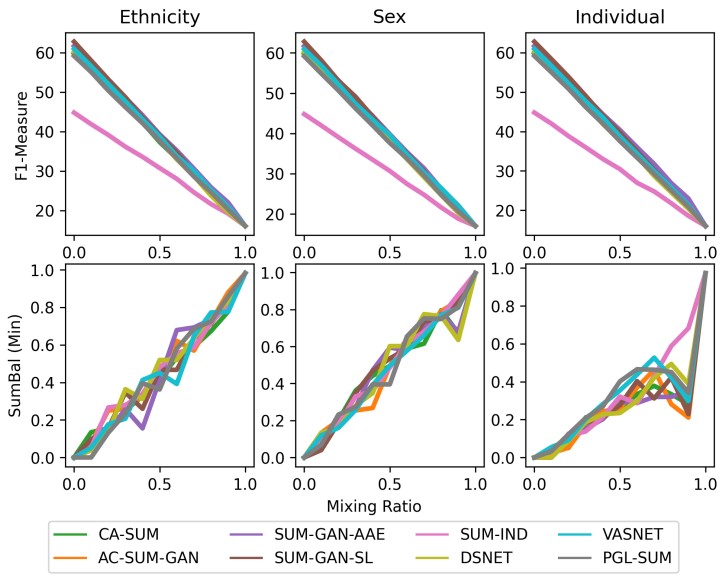

Figure 12: Results for the mixing sampling strategy for Split #3.

standard procedure in all models' codes). Since all methods use a common evaluation code, which involves the Knapsack algorithm to generate summaries based on frame importance scores and the $F_1$ score evaluation, we created a unified evaluation script for reporting the average $F_1$ scores. This script takes the predicted summaries (post-Knapsack) from models and the ground truth user summaries (from annotations) from the h5 dataset as input. We also developed a unified script for evaluating SumBals, which takes as inputs the predicted summaries and user summaries, similar to the $F_1$ score evaluation. The SumBal evaluation also requires the fairness labels, which are numpy binaries provided for all individuals/entities in the frames for each video. Please follow the README on our github or instructions on our webpage for detailed instructions on how to train and evaluate all models.

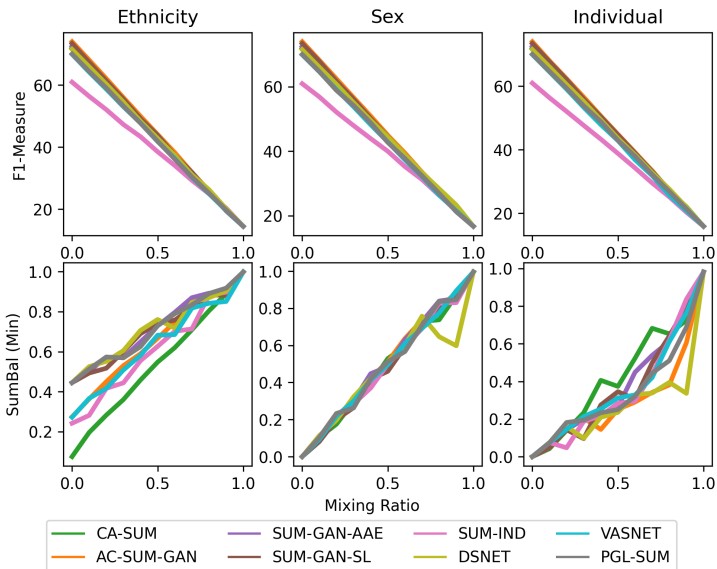

Figure 13: Results for the mixing sampling strategy for Split #4.

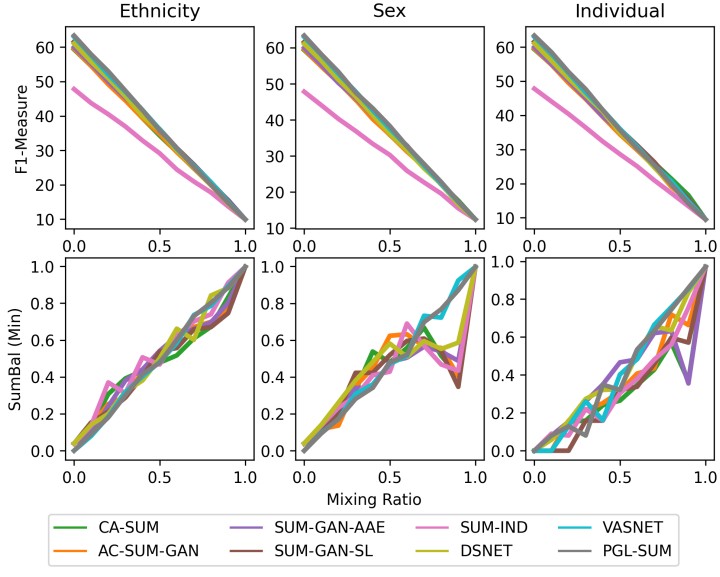

Figure 14: Results for the mixing sampling strategy for Split #5.

## D.4 Videos in *FairVidSum*

*FairVidSum* currently consists of the following YouTube videos:

- Video #1:
https://www.youtube.com/watch?v=YtoSLVGc_vw
- Video #2:
https://www.youtube.com/watch?v=uKSxrQHpCqo
- Video #3:
https://www.youtube.com/watch?v=gAQggxc_aYw
- Video #4:
https://www.youtube.com/watch?v=1rS8fFbW57o&ab_channel=MSNBC

- Video #5:
`https://www.youtube.com/watch?v=6-cVocxHjFw&ab_channel=ArtofCharm`
- Video #6:
`https://www.youtube.com/watch?v=ryHAIdm_eXo&ab_channel=TalesFromSYLRanchDARKROOM`
- Video #7:
`https://www.youtube.com/watch?v=aCDItvntHFE&ab_channel=MyHartEnt`
- Video #8:
`https://www.youtube.com/watch?v=UgYy2maGCU4`
- Video #9:
`https://www.youtube.com/watch?v=ExJZAegsOis`
- Video #10:
`https://www.youtube.com/watch?v=DHHbgnFVyWQ&ab_channel=Impetos`
- Video #11:
`https://www.youtube.com/watch?v=R9jB4JOi6gU&ab_channel=ILVOLOSIM`
- Video #12:
`https://www.youtube.com/watch?v=naIkpQ_cIt0&ab_channel=ESLLearning`
- Video #13:
`https://www.youtube.com/watch?v=JSLhP8i-5U0&ab_channel=ESLLearning`
- Video #14:
`https://www.youtube.com/watch?v=7dixfiGekhE&ab_channel=GOBALDAILYNEWSUSA`
- Video #15:
`https://www.youtube.com/watch?v=bYzH3zP7iDg&ab_channel=MTVUK`
- Video #16:
`https://www.youtube.com/watch?v=BNcFuU23CkQ&ab_channel=SilviuTolu`
- Video #17:
`https://www.youtube.com/watch?v=bJOhf3_fQ-c&ab_channel=MrHG94`
- Video #18:
`https://www.youtube.com/watch?v=t4fAq17jZ-A`
- Video #19:
`https://www.youtube.com/watch?v=C8gHpnZ0BrY`
- Video #20:
`https://www.youtube.com/watch?v=7PnBQwI_M2A`
- Video #21:
`https://www.youtube.com/watch?v=KJ804YQEaVc`
- Video #22:
`https://www.youtube.com/watch?v=rM1jUXSFuls`
- Video #23:
`https://www.youtube.com/watch?v=3eYKfiOEJNs`
- Video #24:
`https://www.youtube.com/watch?v=4wU_LUjG5Ic`
- Video #25:
`https://www.youtube.com/watch?v=akI8YFjEmUw`
- Video #26:
`https://www.youtube.com/watch?v=eQu1rNs0an0`
- Video #27:
`https://www.youtube.com/watch?v=iVt07TCkFM0`
- Video #28:
`https://www.youtube.com/watch?v=jcoYJXDG9sw`
- Video #29:
`https://www.youtube.com/watch?v=JgHubY5Vw3Y`
- Video #30:
`https://www.youtube.com/watch?v=Se3oxnaPsz0`
- Video #31:
`https://www.youtube.com/watch?v=sTEELN-vY30`

- Video #32:
`https://www.youtube.com/watch?v=LRw_obCPUt0`
- Video #33:
`https://www.youtube.com/watch?v=RBCABdttQmI`
- Video #34:
`https://www.youtube.com/watch?v=E11zDS9XGzg`

### D.5 Dataset Collection

We performed the frame labeling process using the Kili labeling platform [6]. Each user participating in the labeling task was instructed to read the video title and watch the video on mute first. For annotation purposes, each user received frames from 22 videos, which were downsampled to 1/2 frames per second. The users were asked to assign an importance score ranging from 1 (least important) to 5 (most important) to each frame. To prevent any potential bias, frames from every video were shuffled before being presented to the users for annotation.

### D.6 Dataset Details

All the models included in the benchmark provide the datasets (*SumMe* and *TVSum*) in structured h5 format, which can be accessed either through their respective github repositories or download links, and also provide their splits.json files used for training and evaluating models. In a similar manner, we have included the FairVidSum dataset in the fvs.h5 file, along with the corresponding fvs_splits.json file used for our benchmarks. Both files are already provided in the repository, and we also provide the dropbox download links to them.

We also provide fair_npy_data/ , which contains all fairness labels and data required for SumBal evaluations and generating summaries using FVS-LP. fair_npy_data/ contains three subfolders, sex/ , eth/ , and ind/ which further contain corresponding fairness labels for each video in our dataset in numpy binary (.npy) format. There is a corresponding .npy file associated with each video (named video_<vid_num>.npy). These .npy files are structured as numpy arrays, with rows representing the frame index, and columns representing the protected groups. For each group, the value of 0 in the numpy array indicates that the corresponding protected group is not present in that frame, while a value of 1 indicates its presence.

The *TVSum* videos (Vid. #23 - #34) were solely labeled for fairness. The h5 dataset was taken directly from the benchmarking model repositories (eccv16_dataset_tvsum_google_pool5.h5), and we simply extracted the chosen 12 videos. Hence, the *TVSum* videos adhere to their downsampling rate (2 frames per second), user annotations, frame features, etc. We append the TVSumm 12 video h5 dataset to our fvs.h5 (containing 22 videos) to create a standard video summarization dataset of 34 videos.

### D.7 Dataset License

The *FairVidSum* dataset is released under a *CC-BY-SA* license. Please refer to `https://creativecommons.org/licenses/by-sa/4.0/` for license details.

### D.8 Videos for Dataset Extension

We plan to add the following videos to the next version of the dataset:

1. `https://www.youtube.com/watch?v=J4SnZcLgX6c`
2. `https://www.youtube.com/watch?v=uNUSI_9lXGg`
3. `https://www.youtube.com/watch?v=tGIhpc8SJK0`
4. `https://www.youtube.com/watch?v=OVvPntIAcsQ&ab_channel=RealMadrid`
5. `https://www.youtube.com/shorts/whdf1kp2Zrs`
6. `https://www.youtube.com/shorts/lbNRKjGHOvg`

---

[6]`https://kili-technology.com/`

7. https://www.youtube.com/shorts/NIlWmDrfxac
8. https://www.youtube.com/shorts/n4e9PiiFOms
9. https://www.youtube.com/shorts/IDJivGVDcos
10. https://www.youtube.com/shorts/PNZdp8Mt6i4
11. https://www.youtube.com/shorts/ytvk5hUnmow
12. https://www.youtube.com/shorts/L7Ag06cp9V8
13. https://www.youtube.com/shorts/BIFicBzErMk
14. https://www.youtube.com/shorts/Ld_Qoxka9jk
15. https://www.youtube.com/shorts/EL4nlb_yuYE
16. https://www.youtube.com/shorts/rY8NyCKq-ic
17. https://www.youtube.com/watch?v=pZLVKmHI8Aw
18. https://www.youtube.com/watch?v=WH71R4PkvmQ
19. https://www.youtube.com/watch?v=uEwCHTSuZMO
20. https://www.youtube.com/watch?v=3EsqpF-W_wQ
21. https://www.youtube.com/watch?v=XJvvzBen91w
22. https://www.youtube.com/watch?v=5bDQE8EzcFk
23. https://www.youtube.com/watch?v=Fh16FBHYyRM
24. https://www.youtube.com/watch?v=Iy7drPXqzps
25. https://www.youtube.com/watch?v=aO-Wz-H_EU4
26. https://www.youtube.com/watch?v=-Dsbg-kCP0g

# E  Ethics Statement

Our paper studied the important and novel problem of fair video summarization. As individuals and groups (based on sensitive attributes) can be directly impacted by automated video summarization, it is important to recognize the importance of ethical considerations in the development and deployment of such systems. Without sensitive attribute information available for video summarization, it is not possible for methods being developed to adhere to the principles of fairness, accountability, and transparency as part of their research goals. Therefore, our work primarily aims to bridge this gap, and address potential biases and discriminatory effects in the video summarization task. However, it is important to also advocate for the responsible use of fair video summarization, and ensure that the provided data and benchmarks are only used to enhance the fairness of existing methods or propose novel fair variants to current models.

# F  Possible Directions for Future Work

There are multiple directions for possible future work:

## F.1  Relaxing Knowledge of Protected Groups

Future work can be proposed that relaxes the assumption on knowledge of protected group annotations, either partially or fully, or can even aim to predict them and then ensure fairness. All these problem settings give rise to a diverse set of non-trivial problem formulations. For instance, methods that assume probabilistic group memberships (such as when a classifier is used to predict the group) would differ significantly from methods that assume partial knowledge of certain groups.

## F.2  Generalized Methods for Different Definitions of Fairness

Currently, video summarization methods differ significantly from each other– some employ LSTMs, others employ GANs, etc. However, for future research on fair video summarization, we believe a more generalized and unified direction for methods can be studied. For example, approaches can be proposed that take an existing video summarization model as input and make it fair for any general fairness definition provided at run-time. Moreover, new fairness definitions for fair video summarization specific to certain applications, can be motivated.

### F.3 Pre-processing, In-processing, and Post-processing Approaches for Fairness

Fairness methods can be pre-processing (transform the input data space to ensure fairness), in-processing (fair model variants), or post-processing (modifying the output summaries to make them fair) based. All three modalities are important to study for future work, as different video summarization pipelines have different data pipeline requirements. Furthermore, it is important to understand how each type of method differs, and whether some are better at ensuring fairness than others.

