# OpenReview forum: "Towards Fair Video Summarization"
_TMLR — Accepted by TMLR_

### Review · Reviewer_ZZ3b · 2023-09-23

**Summary Of Contributions:**

This work is the first to address the concept of fairness in video summarization. The authors provide an analytical definition of fair video summarization and a novel metric called SumBal to measure the fairness in video summaries. To benchmark existing video summarization models, the authors newly collect the dataset, FairVidSum, which is designed similarly to existing datasets, TVSum and SumMe, but also with individual- and group-level fairness annotations. The experiments conducted on FairVidSum clearly showcase the fairness-utility gap between existing video summarization models and the proposed baseline, FVS-LP, that are optimized only for utility and fairness, respectively.

**Audience:**

Yes

**Broader Impact Concerns:**

The authors adequately addressed the ethical implications of the work.

**Claims And Evidence:**

Yes

**Requested Changes:**

- Please add the discussion about the fairness in summarizing videos of different domains and settings that the collected benchmark FairVidSum does not cover.
- Please fix the mismatch in descriptions of the frame sampling rate for important score annotation: the sampling rate is stated as 2 fps in Section 4.1 while 0.5 fps in Appendix D.5.

**Strengths And Weaknesses:**

Strengths
+ Overall, the paper is well-written and easy to follow.
+ The authors clearly define the new task, fair video summarization, and provide all for the development of fair video summarization models, including the new evaluation metric and the benchmark dataset.
+ The experimental results clearly show that existing video summarization models do not account for fairness, at least according to the evaluation metric, SumBal.

Weaknesses
- The new benchmark, FairVidSum, contains videos of a few selected categories such as news, interviews and panel discussion so that each video features a very few individuals in a restricted setting (at most 16 in Video #33). The paper lacks the discussion about the fairness in summarizing videos that feature much more individuals and diverse settings, such as concert or large lecture videos featuring thousands of people.

---

### Review · Reviewer_5vEw · 2023-09-30

**Summary Of Contributions:**

This paper as is titled takes a step towards fair video summarization. They collect a dataset, propose a metric and an unsupervised video summarization baseline.
Video summarization in this work (and other related words) is defined as the task of selecting a sub-set of the video frames from the whole video from sparsely sampled frames with a low frame-rate. In this task no audio is used. Video summarization is usually evaluated using the F-measure.
This work introduces the FairVidSum dataset. The dataset consists of 22 new videos from Youtube and 12 videos from the TVSum dataset. The nature of the videos are "people talking" like podcasts, interviews, etc.
In this work a fair summarizer would select frames from the video that respects and contains the same proportion of protected groups (such as an ethnicity) as it is present in the input video itself.
The proposed evaluation metric SumBal is measuring the most violated proportion from a set of protected groups in the video summarization output.
The proposed baseline is an supervised approach for video summarization that given the proportions of the protected groups in the input video and their presence in each frame, is able to output a summarization that maximally satisfies the fairness goal using a linear program without any other utility or accuracy in its objective.

**Audience:**

Yes

**Broader Impact Concerns:**

I do not have any concerns.

**Claims And Evidence:**

Yes

**Requested Changes:**

- Improve the baseline to incorporate utility/accuracy of the summarization alongside fairness.
- Increase the size of the datset.
- Treatment of the video should be as a video not as a collection of images sampled from a video after removing the audio.

**Strengths And Weaknesses:**

The paper very well written. The problem, related work and the contributions as well explained and presented clearly.

It seems that the video summarization problem as defined in the literature and in this work is not really concerned with the "video" itself, but what it tries to do is to select a subset from the inputs. Firstly, the audio of the video is removed. Secondly, the frames are randomly selected at a low frame-rate without attention to the content. Finally, during the annotation process the frames of the video are shuffled.
I think that selecting a frame for video summarization should be concerned with the surrounding frames and the audio specially in "people talking" setting, it is important to know what is being said and decision cannot be made solely based on the visual features of a single frame.

Furthermore the problem of unsupervised fair summarization seems unrealistic as it assumes that the ground truth group memberships `H` are available. In a realistic setting, given a new test video, `H` has to be predicted but this was not studied in this work.

The proposed baseline seems too rudimentary. There is no hyper-parameter to control for a trade-off between the utility and fairness.

---

### Review · Reviewer_EwqW · 2023-10-02

**Summary Of Contributions:**

This paper proposes a dataset and evaluation metric for fairness in video summarization. They explore many existing video summarization methods using these components. The paper proposes evaluating fairness of a summarization by measuring the proportion of the summary each entity takes up compared to the proportion of the video each entity takes up. The dataset is collected similar to others, but annotated by humans counting each entity.

**Audience:**

Yes

**Broader Impact Concerns:**

It is discussed in the paper.

**Claims And Evidence:**

Yes

**Requested Changes:**

It would be helpful to add more insights in the results section and more experiments studying different aspects of the models, what makes them fair or unfair, etc.

**Strengths And Weaknesses:**

The main strengths of the paper are proposing a new dataset and evaluation metric for studying fairness. There is a lot of analysis on the dataset and the distribution of people, attributes, etc. in the dataset. There is good discussion on the limitations of the dataset.

The main weakness is the evaluation metric is pretty simple. It is just calculating the percent of time an entity is in the summary vs. the source material. This will provide some insight, but is also limited to only datasets where these annotations are available, and any biases in those datasets.

The evaluations are done by training the methods on the new dataset. Have you considered evaluating pretrained models directly on this dataset without any further training?

There's not much analysis or insight in the results. Why are some methods better than others?

---

### Review · Reviewer_8FYH · 2023-10-04

**Summary Of Contributions:**

This paper proposes a fair video summarization task. They build a FairVidSum dataset and SumBal metric for quantifying fairness. They also provide a fair-only baseline called FVS-LP.

**Audience:**

Yes

**Claims And Evidence:**

Yes

**Requested Changes:**

See Weaknesses.

**Strengths And Weaknesses:**

Strengths:
1. The target issues of the paper are meaningful and worth exploring. This submission gives a valuable implementation of such an idea and presents good results.
2. The paper is generally well-written, clearly structured, and quite easy to follow.

Weaknesses:
1.  The motivation is not so clear. In Section 3.2, the authors claim that all persons in the video should be represented in approximately the same proportions in the generated summary as they appear in the entire video.  What if some people are not important for the video? For example, although the main character's frames are less than sub-character, we still focus on the main character. In this situation, fairness is a conflict of importance.
2. The number of videos in the dataset should be more.
3. The author does not provide suggestions for further improvements of methods in their dataset.

---

> ### Comment · Action_Editors · 2023-10-24
> **official recommendation**
>
> Dear Reviewer 8FYH,
>
> Much appreciated for reviewing the paper.  Could you please submit the final recommendation to this paper? thanks!
>
> best,
>
> AE

---

### Decision · Action_Editor_pKx5 · 2023-11-21

**Recommendation:** Accept with minor revision

**Comment:**

These manuscripts underwent review by four experts, with three providing the final recommendations. The cumulative scores present a diverse perspective, with two experts advocating for acceptance, and one reviewer proposing rejection.

Most reviewers acknowledged the novelty of the fair video summarization problem, praised the quality of the gathered dataset, and commended the paper's writing. Reviewer 5vEw identified certain weaknesses in the proposed baseline, the nature of the addressed task, and the dataset size. These comments are generally insightful, and the Associate Editor (AE) values the feedback from Reviewer 5vEw.

Video summarization has been extensively explored in both the Multimedia and vision communities, with various methods developed. This paper focuses on addressing the fairness issue within recent learning-based approaches. The authors have made commendable contributions in this regard. According to the TMLR evaluation criteria, the authors' claims align well with the presented evidence. However, the AE notes the abundance of existing literature on video summarization, suggesting that the authors should discuss the concerns raised by Reviewer 5vEw—specifically regarding baselines and datasets—in the related work and method sections. Consequently, the AE recommends acceptance with minor revisions.

**Audience:**

This paper may be interested to the researchers in machine learning and vision communities. It may be particularly inspiring to those audiences working for some real-world applications, as fairness is indeed an important topic.

**Claims And Evidence:**

This paper studies the fairness of video summarization, particularly in the learning paradigm. This works gives a relatively valuable implementation of such an idea and presents good results.

The claims made in the submission are generally well supported by the evidence. The paper is generally well-written, clearly structured, and quite easy to follow.

---

> ### Author Response · Authors · 2023-12-02
> **Thank you to AE and Reviewers**
>
> Dear AE,
>
> Thank you for all your efforts in managing our submission and for the positive feedback. We are very grateful to you and to all the reviewers for helping strengthen our work and contributions. We have made the requested revisions, by mentioning the difference between classical and learning-based video summarization approaches in the Introduction (right before our listed contributions) and by adding a new Related Works subsection on Classical Video Summarization, in which we discuss classical methods in more detail. Here, we also provide distinctions between learning-based approaches and classical approaches, and mention that our work primarily focuses on learning-based approaches.
>
> Thank you once again.
>
> Kind Regards,
>
> Authors.